# Robust Deep Signed Graph Clustering via Weak Balance Theory

## Abstract

Signed graph clustering is a critical technique for discovering community structures in graphs that exhibit both positive and negative relationships. We have identified two significant challenges in this domain: i) existing signed spectral methods are highly vulnerable to noise, which is prevalent in real-world scenarios; ii) the guiding principle "an enemy of my enemy is my friend", rooted in *Social Balance Theory*, often narrows or disrupts cluster boundaries in mainstream signed graph neural networks. Addressing these challenges, we propose the Deep Signed Graph Clustering framework (DSGC), which leverages *Weak Balance Theory* to enhance preprocessing and encoding for robust representation learning. First, DSGC introduces Violation Sign-Refine to denoise the signed network by correcting noisy edges with high-order neighbor information. Subsequently, Density-based Augmentation enhances semantic structures by adding positive edges within clusters and negative edges across clusters, following *Weak Balance* principles. The framework then utilizes *Weak Balance* principles to develop clustering-oriented signed neural networks to broaden cluster boundaries by emphasizing distinctions between negatively linked nodes. Finally, DSGC optimizes clustering assignments by minimizing a regularized clustering loss. Comprehensive experiments on synthetic and real-world datasets demonstrate DSGC consistently outperforms all baselines, establishing a new benchmark in signed graph clustering. The code is provided in https://anonymous.4open.science/r/DSGC-C05C/.

## Keywords

Representation learning; Balance theory; Signed graph clustering

**ACM Reference Format:**
Anonymous Author(s). 2018. Robust Deep Signed Graph Clustering via Weak Balance Theory. In *Proceedings of conference title (Conference acronym 'XX)*. ACM, New York, NY, USA, 13 pages. https://doi.org/X.X

## 1 Introduction

Deep graph clustering has emerged as a pivotal technique for uncovering underlying communities within complex networks. However, existing methods [2, 28, 32, 35, 37] predominantly target unsigned graphs, which represent relationships solely with "non-negative" edges and inherently fail to capture conflicting node interactions, such as friendship versus enmity, trust versus distrust, and approval versus denouncement. Such dynamics are commonplace in social networks and can be effectively modeled by signed graphs

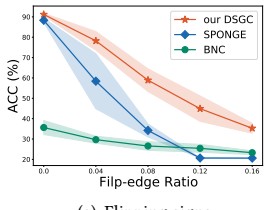 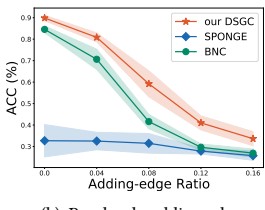

(a) Flipping signs  (b) Randomly adding edges

**Figure 1: Effects of different perturbations, including flipping signs and randomly adding negative edges, on the clustering performance of popular spectral methods in signed graphs.**

that incorporate both positive and negative edges. Although significant work has studied link prediction tasks in deep signed graphs [8, 16, 17, 24, 38, 43], deep signed graph clustering remains substantially unexplored. In this paper, we aim to develop a deep signed graph clustering method that enhances the robustness of graph representations, facilitating more distinctive clusters and better reflecting the intricate relationships within signed graphs.

Signed graph clustering is broadly applied in the analysis of social psychology [7, 21, 30], biologic gene expressions [11, 33], etc. Recent studies have predominantly focused on spectral methods [5–7, 30], which design various Laplacian matrices specific to a given network to derive node embeddings, aiming to find a partition of nodes that maximizes positive edges within clusters and negative edges between clusters. However, these methods are vulnerable to random noise, a common challenge in real-world scenarios. For instance, on shopping websites, the signed graph encoding user-product preferences often includes noisy edges, typically when customers unwillingly give positive ratings to items in exchange for meager rewards or coupons. Fig. 1 illustrates the significant impact of noise on signed spectral methods like BNC and SPONGE [5, 6]. As perturbation ratios increase, which indicates a higher percentage of randomly flipped edge signs or inserted negative edges in a synthetic signed graph with five clusters, these methods suffer a sharp decline in clustering accuracy. Therefore, denoising the graph structure is essential to enhance robust representation learning in deep signed clustering.

Furthermore, the investigation on deep signed graph neural networks (SGNNs) reveals that existing SGNNs — which are mostly developed for link prediction [8, 16, 17, 24, 38, 43] — do not adapt well to signed clustering. Specifically, mainstream SGNNs models typically leverage principles from the well-established *Social Balance Theory* [14] (or Balance Theory) to design their messaging-passing aggregation mechanisms, including the classical principle "*an Enemy of my Enemy is my Friend (EEF)*", "*a Friend of my Friend is my Friend (FFF)*", "*an Enemy of my Friend is my Enemy (EFE)*". However, "*EEF*" implies an assumption that a given signed network has only 2 clusters, which is not directly applied to signed graphs with $K$ ($K > 2$) clusters. Specifically, as illustrated in Fig. 2, "*EEF*"

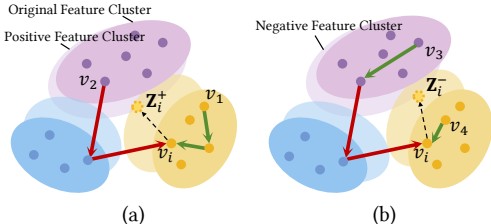

(a)                    (b)

**Figure 2: Illustration of "*an Enemy of my Enemy is my Friend (EEF)*" narrowing cluster boundaries. Aggregating positive (/ negative) neighbor** $2$ **(/** $3$**) causes** $Z_i^+$ **(/** $Z_i^-$**) mapped far from its clusters or even cross the boundary, where positive (/ negative) neighbors** $2$ **(/** $3$**) are defined by "*EEF*".**

can narrow cluster boundaries, leading to more nodes being located at the margins of clusters, which makes it difficult to assign them to the correct clusters and thus results in poor performance. For example, node $v_i$ aggregates its positive neighbor $v_2$ (recognized by "*EEF*" but inconsistent to the real semantic relationship in clusters), which causes its positive representation $Z_i^+$ mapped closer to the cluster of node $v_2$, thus leading to narrowed or even overlapped cluster boundaries. In contrast, *Weak Balance Theory* [7] (or Weak Balance), introducing a new principle, "*an enemy of my enemy might be my friend or enemy*", can generalize Balance Theory to $K$-way ($K > 2$) clustering situation but remains underexplored.

To address these challenges, we propose eep Signed Graph Clustering (DSGC) for $K$-way clustering, designed to enhance representations' robustness against noisy edges and reduce the impact of the ill-suited principle on cluster boundaries. DSGC first introduces the Signed Graph Rewiring module (SGR) in the preprocessing stage for denoising and graph structure augmentation. SGR provides two rewiring strategies, including *Violation Sign-Refine*, which can identify and correct noisy edges with long-range neighbor relationships, and *Density-based Augmentation*, which follows Weak Balance principles to insert new positive edges to increase positive density within clusters and negative edges to increase negative density across clusters. Such refined graph topology can promote signed encoders to enhance the robustness of node representations. DSGC then constructs a clustering-oriented signed neural network that utilizes Weak Balance. This helps design clustering-specific neighbor aggregation mechanism for enhancing the discrimination among node representations, specifically for nodes with negative edges to widen cluster boundaries. Finally, DSGC designs a $K$-way clustering predictor that optimizes a non-linear transformation function to learn clustering assignments. This framework is designed to refine the clustering process by correcting noisy edges and enhancing the discriminative capability of node representations, ultimately leading to more accurate clustering outcomes.

Overall, our major contributions are as follows:

- We develop DSGC, the first Deep Signed Graph Clustering framework, by leveraging Weak Balance Theory.
- We design two graph rewiring strategies to denoise and augment the overall network topology.

- We propose a task-oriented signed graph encoder to learn more discriminative representations, particularly for nodes connected by negative edges.
- Extensive experiments on synthetic and real-world datasets demonstrate the superiority and robustness of DSGC.

## 2 Related Work

In this section, we succinctly review existing studies for signed graph neural networks and signed graph clustering.

**Signed Graph Neural Networks (SGNNs)** , which maps nodes within a signed graph to a low-dimensional latent space, has increasingly facilitated a variety of signed graph analytical tasks, including node classification [29], signed link prediction [18, 40, 42], node ranking [13, 34], and signed clustering [5, 6, 15, 21, 36]. Most works of signed graph center around integrating *Social Balance Theory* to signed convolutions into Graph Neural Networks (GNNs). As the pioneering work, SGCN [9] adapts unsigned GNNs for signed graphs by aggregating and propagating neighbor information with Balance Theory. Thereafter, other work has integrated additional social-psychological theories. [4] appends the status theory, which is applicable to directed signed networks, interpreting positive or negative signs as indicators of relative status between nodes. SiGATs [16], which extends Graph Attention Networks (GATs) to signed networks, also utilizes these two signed graph theories to derive graph motifs for more effective message passing. SiNEs [39] proposes a signed network embedding framework guided by the extended structural balance theory. SGDNET [19] leverages a random walk technique specifically tailored for signed graphs, effectively diffusing hidden node features in line with Social Balance Theory. GS-GNN [25] applies a dual GNN architecture that combines a prototype-based GNN to process positive and negative edges to learn node representations. SLGNN [23] especially design low-pass and high-pass graph convolution filters to capture both low-frequency and high-frequency information from positive and negative links.

**Signed Graph Clustering.** The study of signed graph clustering has its roots in Social Balance Theory [3], which is equivalent to the 2-way partition problem in signed graphs [30]. Building upon this foundational concept, [21] propose a signed spectral clustering method that utilizes the signed graph Laplacian and graph kernels to address the 2-way partition problem. However, [41] argues that community detection in signed graphs is equivalent to identifying $K$-way clusters using an agent-based heuristic. The **Weak Balance Theory** [7] relaxes balance theory to enable $K$-way clustering. Following [21], [5] proposed the "Balanced Normalized Cut (BNC)" for $K$-way clustering, aiming to find an optimal clustering assignment that minimizes positive edges between different clusters and negative edges within clusters with equal priority. SPONGE [6] transforms this discrete NP-hard problem into a continuous generalized eigenproblem and employs LOBPCG [20], a preconditioned eigensolver, to solve large positive definite generalized eigenproblems. In contrast to the above $K$-way complete partitioning, [36] targets detecting $K$ conflicting groups in a signed network, allowing other nodes to be neutral regarding the conflict structure in search. This conflicting-group detection problem can be characterized as

the maximum discrete Rayleigh's quotient problem and solved by two spectral methods.

While GNNs have been extensively applied to unsigned graph clustering [2, 28, 32, 35, 37], their adoption in signed graph clustering remains overlooked. A notable exception is the Semi-Supervised Signed NETwork Clustering (SSSNET) [15], which simultaneously learns node embeddings and cluster assignments by minimizing the clustering loss and a Cross-Entropy classification loss. In contrast, our work develops an unsupervised method for signed graph clustering, eliminating the reliance on ground truth labels.

## 3 Preliminaries

### 3.1 Notations

We denote an undirected signed graph as $\mathcal{G} = \{\mathcal{V}, \mathcal{E}, \mathbf{X}\}$, where $\mathcal{V} = \{v_1, v_2, \ldots, v_n\}$ is the set of nodes, $\mathcal{E}$ is the set of edges, and $\mathbf{X} \in \mathbb{R}^{|\mathcal{V}| \times d_0}$ is the $d_0$-dimensional node attributes. Each edge $e_{ij} \in \mathcal{E}$ between $v_i$ and $v_j$ can be either positive or negative, but not both. $\mathbf{A}$ is the adjacency matrix of $\mathcal{G}$, where $\mathbf{A}_{ij} = 1$ if $v_i$ has a positive link to $v_j$; $\mathbf{A}_{ij} = -1$ if $v_i$ has a negative link to $v_j$; $\mathbf{A}_{ij} = 0$ otherwise. The signed graph is conceptually divided into two subgraphs sharing the common vertex set $\mathcal{V}$: $\mathcal{G} = \{\mathcal{G}^+, \mathcal{G}^-\}$, where $\mathcal{G}^+ = \{\mathcal{V}, \mathcal{E}^+\}$ and $\mathcal{G}^- = \{\mathcal{V}, \mathcal{E}^-\}$ contain all positive and negative edges, respectively. Let $\mathbf{A}^+$ and $\mathbf{A}^-$ be the adjacency matrices of $\mathcal{G}^+$ and $\mathcal{G}^-$ with $\mathbf{A} = \mathbf{A}^+ - \mathbf{A}^-$, where $\mathbf{A}_{ij}^+ = max(\mathbf{A}_{ij}, 0)$ and $\mathbf{A}_{ij}^- = -min(\mathbf{A}_{ij}, 0)$.

### 3.2 Relaxation of Social Balance

Balance and Weak Balance Theories, essential for signed graph clustering, are briefly explained here; more details are in Appx. A.

**Balance Theory** [14] consists of four fundamental principles: "*the friend of my friend is my friend*", "*the enemy of my friend is my enemy*", "*the friend of my enemy is my enemy*", and "*the enemy of my enemy is my friend (EEF)*". A signed network is balanced if it does not violate these principles. Theoretically, the Balance Theory is equivalent to 2-way clustering on graphs [30].

**Weak Balance Theory** [7] relaxes Balance Theory to accommodate $K$-way clustering, by replacing the "*EEF*" principle with "*the enemy of my enemy might be my enemy (EEE)*". This principle allows nodes in a triangle to belong to three different clusters, e.g., the blue triangle in Fig. 10 (b), thus relaxing Social Balance Theory. The partition $\{C_1, \ldots, C_K\}$ of a signed graph $\mathcal{G}$ satisfying either theory can be uniformly defined as the following conditions:

$$\begin{cases} \mathbf{A}_{ij} > 0 & (e_{ij} \in \mathcal{E}) \cap (v_i \in C_k) \cap (v_j \in C_k) \\ \mathbf{A}_{ij} < 0 & (e_{ij} \in \mathcal{E}) \cap (v_i \in C_k) \cap (v_j \in C_l)(k \neq l) \end{cases}, \quad (1)$$

where $\mathbf{A}_{ij}$ is the weight of edge $e_{ij}$ and $0 < k, l < K$.

### 3.3 Problem Definition

This paper aims to leverage the capabilities of deep representation learning to enhance robust graph signed clustering. Unsupervised **Deep Signed Graph Clustering** is formally defined below.

PROBLEM 1. *Given a signed graph* $\mathcal{G} = \{\mathcal{V}, \mathcal{E}, \mathbf{X}\}$, *deep signed graph clustering is to train a function* $f(\mathbf{A}, \mathbf{X}) \longrightarrow \mathbf{Z}$ *that transforms each node* $v \in \mathcal{V}$ *into a low-dimensional vectors* $\mathbf{Z}_v \in \mathbb{R}^d$. *It aims to optimize a partition to divide all nodes* $\{\mathbf{Z}_i\}_{i=1}^{|\mathcal{V}|}$ *into K disjoint*

clusters $\mathcal{V} = C_1 \cup \cdots \cup C_K$, *by minimizing a signed clustering loss objection that makes as many as positive edges exist within clusters and as many as negative edges exist across clusters.*

## 4 Methodology

As illustrated in Fig. 3, DSGC consists of 4 major components, including Violation Sign-Refine and Density-based Augmentation for graph rewiring, signed clustering encoder, and cluster assignment.

### 4.1 Signed Graph Rewiring

In real-world signed graphs, noisy edges(violations)—negative edges within clusters and positive edges across clusters—can disrupt ideal clustering structures. To address this, we propose two graph rewiring methods to enhance clustering integrity: Violation Sign-Refine (VS-R), which corrects the signs of violated edges to align negative and positive edges with the expected inter-cluster and intra-cluster relationships; and Density-based Augmentation (DA), which adds new edges based on long-range interaction patterns to reinforce message passing. Both methods leverage Weak Balance and are used as preprocessing steps to denoise and augment the initial graph topology—specifically, the message-passing matrix.

*4.1.1 Violation Sign-Refine.* To address noisy edges, we utilize high-order neighbor interactions to correct their signs. Based on Weak Balance, we first adapt the definitions of positive and negative walks for $K$-way clustering. Following Social Balance Theory, [10] defines a positive walk as one containing an even number of negative edges and a negative walk as one containing an odd number of negative edges. However, they are not suitable for $K$-way clustering due to the uncertainty brought by the "*the enemy of my enemy might be my enemy or my friend*" principle of Weak Balance. We formally redefine positive and negative walks as follows.

DEFINITION 1. *A walk of length* $l \in \mathbb{N}^+$ *connecting nodes* $v_i$ *and* $v_j$ *is positive if all its edges are positive; it is negative when it contains exactly one negative edge and all other edges are positive.*

Since violations are sparse in graphs, we assume that leveraging higher-order information from longer-range neighbors helps revise the signs of violated edges. Lemma 1 specifies the non-noise score between $v_i$ and $v_j$ w.r.t. the $l$-length positive and negative walks.

LEMMA 1. *For* $v_i, v_j \in \mathcal{V}$ *in a signed graph* $\mathcal{G} = \{\mathcal{V}, \mathcal{E}, \mathbf{X}\}$, *let* $\mu_l^+(i, j)$ *and* $\mu_l^-(i, j)$ *be the number of positive and negative walks with length* $l$ *connecting* $v_i$ *and* $v_j$, *respectively. Then,* $\forall a \in \mathbb{N}$,

$$\mu_l^+(i, j) - \mu_l^-(i, j) = (\mathbf{A}^+)_{ij}^l - \sum_{a=0}^{l-1}((\mathbf{A}^+)^a \mathbf{A}^- (\mathbf{A}^+)^{l-1-a})_{ij}. \quad (2)$$

If we consider all walks up to length $L'$, the non-noise score of the connection between $v_i$ and $v_j$ can be defined:

$$\Gamma_{ij}(L') = \sum_{l=1}^{L'} \alpha_l(\mu_l^+(i, j) - \mu_l^-(i, j)), \quad (3)$$

$$\text{where} \quad \alpha_l = \begin{cases} 1, & l = 1 \\ 1/(l!), & 1 < l < L' \\ 1 - \sum_{l'=1}^{L'-1} 1/(l'!), & l = L' \end{cases} \quad (4)$$

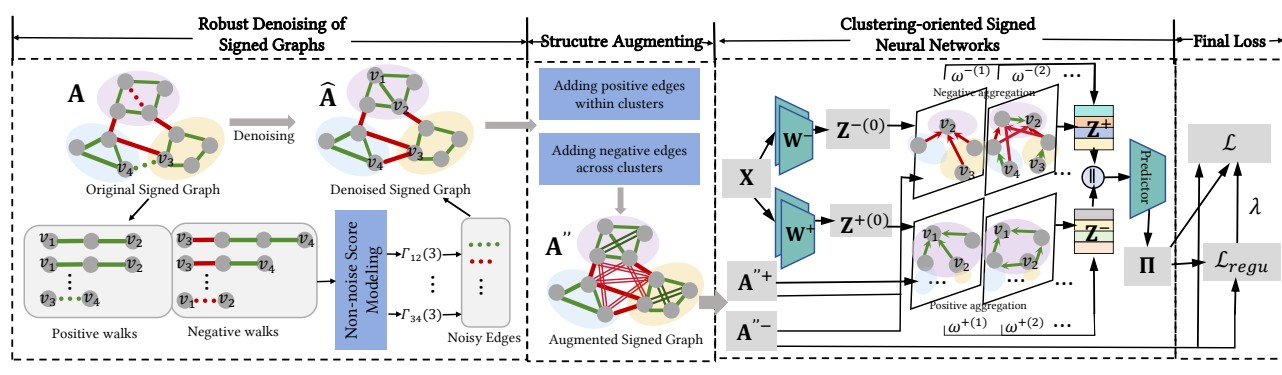

**Figure 3: The overall framework of DSGC. The Violation Sign-Refine first computes non-noise scores to correct the signs of noisy edges. Then, the Density-based Augmentation adds positive edges within clusters and negative edges across clusters. These two rewiring methods generate a new adjacency matrix with reduced noise and enhanced semantic structures. Thereafter, clustering-specific signed convolutional networks can be trained by minimizing the differential clustering loss for learning and strengthening the discrimination among node representations linked negatively.**

Here, $\alpha_l$ decreases with $l$, indicating that shorter walks have more influence. $\Gamma$ is utilized to correct violations as $\Gamma_{ij}$ extracts high-order information from neighbors of $v_i$ and $v_j$ within $L'$-hop. With $\Gamma_{ij}$, we obtain a refined adjacency matrix $\hat{A}$ via the following rules:

$$\hat{A}_{ij} = \begin{cases} 1, & \Gamma_{ij} > \delta^+ \\ A_{ij}, & \delta^- \le \Gamma_{ij} \le \delta^+ \\ -1, & \Gamma_{ij} < \delta^- \end{cases}, \tag{5}$$

where $\delta^+ > 0$ and $\delta^- < 0$ are two thresholds. $v_i$ and $v_j$ are considered *effective friends* when $\Gamma_{ij} > \delta^+$, indicating a positive edge (+); $v_i$ and $v_j$ are considered *effective enemies* when $\Gamma_{ij} < \delta^-$, indicating a negative edge (−); otherwise, the original adjacency entries in $A$ retains. The magnitude $|\Gamma_{ij}|$ represents the confidence level of two nodes being *effective friends* or *enemies*. A larger (resp. smaller) $|\Gamma_{ij}|$ represents a stronger (resp. weaker) positive or negative relationship between $v_i$ and $v_j$. This method refines the adjacency matrix by reinforcing accurate relational signals and reducing the impact of noisy edges, thereby facilitating more effective clustering.

*4.1.2 Density-based Augmentation.* Following the noise corrections made by VS-R, the revised graph, denoted as $\hat{\mathcal{G}} = \{\mathcal{V}, \hat{\mathcal{E}}, X\}$, is processed through Density-based Augmentation to increase the density of positive edges within clusters and negative edges between clusters. The revised adjacency matrices for positive and negative edges, $\hat{A}^+$ and $\hat{A}^-$, are augmented as below:

$$A'^+ = (\hat{A}^+)^{m^+}; \quad A'^- = \sum_{a=0}^{m^-} (\hat{A}^+)^a \hat{A}^- (\hat{A}^+)^{m^- - a}, \tag{6}$$

where $m^+$ and $m^-$ are scalar hyper-parameters indicating the extent of augmentation. The augmented adjacency matrices are:

$$A''^+ = \begin{cases} 1, & A'^+_{ij} > 0, \ i \ne j \\ 0, & A'^+_{ij} = 0, \ i \ne j \\ 0, & A'^+_{ij}, \ i = j \end{cases}; \quad A''^- = \begin{cases} 1, & A'^-_{ij} > 0, \ i \ne j \\ 0, & A'^-_{ij} = 0, \ i \ne j \\ 0, & A'^-_{ij}, \ i = j \end{cases}. \tag{7}$$

If $m^+ = 1$ (resp. $m^- = 0$), no augmentation is performed on $\hat{A}^+$ (resp. $\hat{A}^-$). For $m^+ > 1$, it adds a positive edge between any two nodes connected by a $m^+$-length positive walk (Dfn. 1). For $m^- > 0$, it adds a negative edge between any two nodes connected by a $(m^- + 1)$-length negative walk. This strategy effectively enhances the clustering potential by reinforcing intra-cluster connectivity with positive edges and inter-cluster separations with negative edges. It is particularly effective for a signed graph with few violations.

## 4.2 Signed Clustering Encoder

Signed Graph Convolution Network, our signed graph encoder in DSGC, is tailored for $K$-way clustering. It leverages Weak Balance Theory principles to learn discriminative node representations that signify greater separation between nodes connected by negative edges and closer proximity between those by positive edges [1].

Based on the rewired graph topology defined by $A''^+$ and $A''^-$, we first introduce self-loops to each node using $\tilde{A}^+ = A''^+ + \epsilon^+ I$, $\tilde{A}^- = A''^- + \epsilon^- I$, where $I$ is the identity matrix and $\epsilon^+$ and $\epsilon^-$ are the balance hyperparameters. The adjacency matrices are then normalized as follow: $\bar{A}^+ = (\tilde{D}^+)^{-1} \tilde{A}^+$ and $\bar{A}^- = (\tilde{D}^-)^{-1} \tilde{A}^-$, where $\tilde{D}^+$ and $\tilde{D}^-$ are diagonal degree matrices with $\tilde{D}^+_{ii} = \sum_j \tilde{A}^+_{ij}$ and $\tilde{D}^-_{ii} = \sum_j \tilde{A}^-_{ij}$. We learn $d$-dimensional positive and negative embeddings, $Z^+_i$ and $Z^-_i$, for each node $v_i \in \mathcal{V}$, and concatenate them as the final node representation: $Z_i = \text{CONCAT}(Z^+_i, Z^-_i)$, where $Z^+_i \in \mathbb{R}^{1 \times d}$ and $Z^-_i \in \mathbb{R}^{1 \times d}$ are computed through layers of our signed graph convolution network:

$$Z^+_i = \sum_{l=0}^{L} \omega^{+(l)} Z^{+(l)}_i, \quad Z^-_i = \sum_{l=0}^{L} \omega^{-(l)} Z^{-(l)}_i. \tag{8}$$

$\omega^{+(l)}$ and $\omega^{-(l)}$, shared by all nodes, are layer-specific trainable weights that modulate the contribution of different convolution layers to the final node representation. $L$ is the number of layers

---

[1]This goal is often reflected in the loss function designs in previous work [9, 16, 23–25] for link prediction. However, its importance has been overlooked in signed encoders.

in the neural network. This design allows the encoder to leverage information from different neighborhood ranges. The intermediate representations of all nodes, $\mathbf{Z}^{+(l)} \in \mathbb{R}^{|\mathcal{V}| \times d}$ and $\mathbf{Z}^{-(l)} \in \mathbb{R}^{|\mathcal{V}| \times d}$, can be obtained as

$$\mathbf{Z}^{+(l)} = (\bar{\mathbf{A}}^+)^l \mathbf{Z}^{+(0)}, \tag{9}$$

$$\mathbf{Z}^{-(l)} = \sum_{b=0}^{l-1} (\bar{\mathbf{A}}^+)^b (-\bar{\mathbf{A}}^-)(\bar{\mathbf{A}}^+)^{l-1-b} \mathbf{Z}^{-(0)}, \tag{10}$$

where the superscript $(l)$ and $l$ denote the layer index and power number, respectively. The initial node embeddings, $\mathbf{Z}^{+(0)} \in \mathbb{R}^{|\mathcal{V}| \times d}$ and $\mathbf{Z}^{-(0)} \in \mathbb{R}^{|\mathcal{V}| \times d}$, are derived from the input feature matrix $\mathbf{X} \in \mathbb{R}^{|\mathcal{V}| \times d_0}$ by two graph-agnostic non-linear networks:

$$\mathbf{Z}^{+(0)} = \mathbf{W}_1^+(\sigma(\mathbf{X}\mathbf{W}_0^+)), \tag{11}$$

$$\mathbf{Z}^{-(0)} = \mathbf{W}_1^-(\sigma(\mathbf{X}\mathbf{W}_0^-)), \tag{12}$$

where $\sigma$ is the *ReLU* activation function. $\mathbf{W}_0^+ \in \mathbb{R}^{d_0 \times d}$ and $\mathbf{W}_1^+ \in \mathbb{R}^{d \times d}$ are the trainable parameters of the positive network; $\mathbf{W}_0^- \in \mathbb{R}^{d_0 \times d}$ and $\mathbf{W}_1^- \in \mathbb{R}^{d \times d}$ are that of the negative network. We claim that the positive aggregation function, Eq. (9), can pull the nodes linked by positive walks, thus reducing the intra-cluster variances. Meanwhile, the negative aggregation function, Eq. 10, can push nodes linked by negative walks, thus increasing the inter-cluster variances. We also investigate Weak Balance principles implied in Eq. (9) and Eq. (10), as well as the effect of the minus sign "-" in the term $(-\bar{\mathbf{A}}^-)$ to nodes representations linked negatively and the clustering boundary in App. B.

## 4.3 $K$-way Signed Graph Clustering

With node representations $\mathbf{Z} \in \mathbb{R}^{|\mathcal{V}| \times 2d}$ learned in our encoder, we propose a non-linear transformation to predict clusters.

**Clustering Assignment.** Considering a $K$-way clustering problem, where $K$ is the number of clusters, node $v_i$ is assigned a probabilities vector $\Pi_i = [\pi_i(1), \dots, \pi_i(K)]$, representing the likelihood of belonging to each cluster. $k \in \{1, \dots, K\}$ denotes the index of a cluster and $\sum_{k=1}^K \pi_i(k) = 1$. This probability is computed using a learnable transformation followed by a softmax operation:

$$\pi_i(k) = q_\theta(k|\mathbf{Z}_i) = \frac{\exp(\mathbf{Z}_i \cdot \theta_k)}{\sum_{k'=1}^K \exp(\theta_{k'} \cdot \mathbf{Z}_i)}, \tag{13}$$

where $\theta_k \in \mathbb{R}^{d \times 1}$ is a parameter for cluster $k$ to be trained by minimizing the signed clustering loss. The assignment vectors of all nodes $\{\Pi_i\}_{i=1}^{|\mathcal{V}|}$ form an assignment matrix $\Pi \in \mathbb{R}^{|\mathcal{V}| \times K}$.

**Differential Signed Clustering Loss.** Signed graph clustering aims to minimize violations, which was historically considered as an NP-Hard optimization problem [21] with designed discrete (non-differential) objectives in spectral methods. We transform it into a differentiable format by utilizing a soft assignment matrix $\Pi$ in place of a hard assignment matrix $\mathbf{C}$. Specifically, given that the cluster number $K$ is known, let $\mathbf{C} \in \{0, 1\}^{|\mathcal{V}| \times K}$ be a hard cluster assignment matrix where $\mathbf{C}_{(:,k)}(i) = 1$ if node $v_i$ belong to the cluster $k$; otherwise $\mathbf{C}_{(:,k)}(i) = 0$. The number of positive edges between cluster $k$ and other clusters can be captured by $\mathbf{C}_{(:,k)}^T \mathbf{L}^+ \mathbf{C}_{(:,k)}$ with the positive graph Laplacian $\mathbf{L}^+ = \mathbf{D}^+ - \mathbf{A}^+$. The number of negative edges within cluster $k$ can be measured by

$\mathbf{C}_{(:,k)}^T \mathbf{A}^- \mathbf{C}_{(:,k)}$. So the violations w.r.t. cluster $k$ can be measured by $\mathbf{C}_{(:,k)}^T (\mathbf{L}^+ + \mathbf{A}^-) \mathbf{C}_{(:,k)}$. By replacing the hard assignment $\mathbf{C}_{(:,k)}$ with the soft assignment probability $\Pi_{(:,k)}$, the differential clustering loss is constructed as:

$$\mathcal{L} = \frac{1}{|\mathcal{V}|} \sum_{k=1}^K \Pi_{(:,k)}^T (\mathbf{L}^+ + \mathbf{A}^-) \Pi_{(:,k)} + \lambda \mathcal{L}_{regu}, \tag{14}$$

where $\lambda$ is a hyperparameter, and $\mathcal{L}_{regu}$ is a regularization term computing the degree volume in cluster to prevent model collapse:

$$\mathcal{L}_{regu} = -\frac{1}{|\mathcal{V}|} \sum_{k=1}^K \Pi_{(:,k)}^T \overline{\mathbf{D}} \Pi_{(:,k)}, \tag{15}$$

where $\overline{\mathbf{D}}$ is the degree matrix of $\mathbf{A}$. Minimizing $\mathcal{L}$ equals finding a partition with minimal violations. We iteratively optimize the signed encoder and non-linear transformation by minimizing $\mathcal{L}$.

**Inference stage.** Each node $v_i \in \mathcal{V}$ is assigned to the cluster with the highest probability in its vector $\Pi_i$:

$$s_i = argmax_k \Pi_i, \tag{16}$$

where $s_i \in \{1, \dots, K\}$ is the cluster index for $v_i$. The set of all node cluster assignments, $\{s_i\}_{i=1}^{|\mathcal{V}|}$, is used to evaluate the performance of the clustering approach.

## 5 Experiments

This section evaluates our DSGC model with both synthetic and real-world graphs to address the following research questions. **RQ1:** Can DSGC achieve state-of-the-art clustering performance on signed graphs without any labels? **RQ2:** How does each component contribute to the effectiveness of DSGC? **RQ3:** How does the Violation Sign-Refine (VS-R) impact signed topology structures by correcting noisy edges? **RQ4:** How do the strategies in our signed encoder, specifically abandoning the "*EEF*" principle and the minus sign in term $(-\bar{\mathbf{A}}^-)$, contribute to forming wider clustering boundaries?

## 5.1 Experimental Settings

*5.1.1 Datasets.* Follow SPONGE [6], we evaluate DSGC with a variety of synthetic and real-world graphs: (i) **Synthetic SSBM graphs.** The Signed Stochastic Block Model (SSBM) is commonly used to generate labeled signed graphs [6, 31], parameterized by $N$ (number of nodes), $K$ (number of clusters), $p$ (edge probability or sparsity), and $\eta$ (sign flip probability). This model first sets edges within the same cluster as positive, and edges between clusters as negative. It then models noises by randomly flipping the sign of each edge with probability $\eta \in [0, 1/2)$. Each generated graph can be represented as SSBM $(N, K, p, \eta)$. (ii) **Real-world graphs.** S&P is a stock correlation network from market excess returns during $2003 - 2005$, consisting of $1,193$ nodes, $1,069,319$ positive edges, and $353,930$ negative edges. Rainfall is a historical rainfall dataset from Australia, where edge weights are computed by the pairwise Pearson correlation. Rainfall is a complete signed graph with $306$ nodes, $64,408$ positive edges, and $29,228$ negative edges.

*5.1.2 Baselines.* DSGC is compared against 9 representative signed spectral clustering methods. Five are basic signed spectral methods utilizing various forms of Laplacian matrix: (1) symmetric adjacency matrix $\mathbf{A}^* = \frac{1}{2}(\mathbf{A} + \mathbf{A}^T)$; (2) simple normalized signed Laplacian $\overline{\mathbf{L}}_{sns} = \overline{\mathbf{D}}^{-1}(\mathbf{D}^+ - \mathbf{D}^-\mathbf{A}^*)$; (3) balanced normalized signed Laplacian $\overline{\mathbf{L}}_{bns} = \overline{\mathbf{D}}^{-1}(\mathbf{D}^+ - \mathbf{A}^*)$; (4) signed Laplacian graph $\overline{\mathbf{L}} = \overline{\mathbf{D}} - \mathbf{A}$ with a diagonal matrix $\overline{\mathbf{D}}$; and (5) its symmetrically normalized version $\mathbf{L}_{sym}$ [21]. Two are $K$-way spectral clustering methods: (6) Balanced Normalized Cut (BNC) and (7) Balanced Ratio Cut (BRC) [5]. The last two [6] are two generalized eigenproblem formulations: (8) SPONGE and (9) SPONGE$_{sym}$. Moreover, we compare with 6 state-of-the-art deep unsigned graph clustering methods: (10) DAEGC [37], (11) DFCN [35], (12) DCRN [28], (13) Dink-net [27], (14) DGCLUSTER [1] (15) MAGI [26]. Please refer to App. E for hyperparameters settings and experiment details.

*5.1.3 Evaluation Metrics.* For *Labeled graphs (SSBM)*, Accuracy (ACC), Adjusted Rand Index (ARI) [12], Normalized mutual information (NMI), and F1 score are used as the ground truths of nodes are available. For *Unlabeled graphs (S&P and Rainfall)*, due to the lack of ground truths, clustering quality is visualized by plotting network adjacency matrices sorted by cluster membership. See more detailed settings in App. E.

## 5.2 Overall Performance

To address **RQ1**, we evaluated our DSGC and baselines on a variety of labeled signed graphs generated from four SSBM configurations, including SSBM ($N = 1000$, $K = 5$, $p = 0.01$, $\eta$) with $\eta \in \{0, 0.02, 0.04, 0.06, 0.08\}$, SSBM ($N = 1000$, $K = 10$, $p$, $\eta = 0.02$) with $p \in \{0.01, 0.02, 0.03, 0.04, 0.05\}$, SSBM ($N$, $K = 5$, $p = 0.01$, $\eta = 0$) with $N \in \{300, 500, 800, 1000, 1200\}$, and SSBM ($N = 1000$, $K$, $p = 0.01$, $\eta = 0.02$) with $K \in \{4, 5, 6, 7, 8\}$. The performance of each experiment was measured by taking the average of 5 repeated executions. Table 1 reports the results in ACC and NMI. Appendix F provides the results in ARI and F1 score.

Table 1 shows: (i) *Superior performance:* Our DSGC significantly outperforms all baseline models across all metrics, even though SPONGE and SPONGE$_{sym}$ are known for their effectiveness on such datasets. (ii) *Robustness:* Regardless of whether the graph is dense or sparse ($p$), large or small ($N$), noisy or clean ($\eta$), and the number of clusters is few or many ($K$), DSGC maintains notably superior performance on all 20 labeled signed graphs. (iii) *Comparative analysis:* While deep unsigned clustering methods (DAEGC, DFCN, DCRN, Dink-net, DGCLUSTER MAGI) consistently underperform our DSGC due to the limitation of their capabilities to only handle non-negative edges. DSGC still has a clear advantage, highlighting the effectiveness of its design specifically tailored for signed graph clustering.

## 5.3 Ablation Study

To address **RQ2** and evaluate the contributions of key components of DSGC, we performed an ablation study using labeled signed graphs, including SSBM (1000, 10, $p$, 0.01), SSBM (1000, 5, 0.01, $\eta$), SSBM ($N$, 5, 0.01, 0.01), and SSBM (1000, $K$, 0.01, 0.01). The variants of DSGC tested are: **w/o VS-R** is DSGC without Violation Sign-Refine; **w/o DA** is DSGC without Density-based Augmentation;

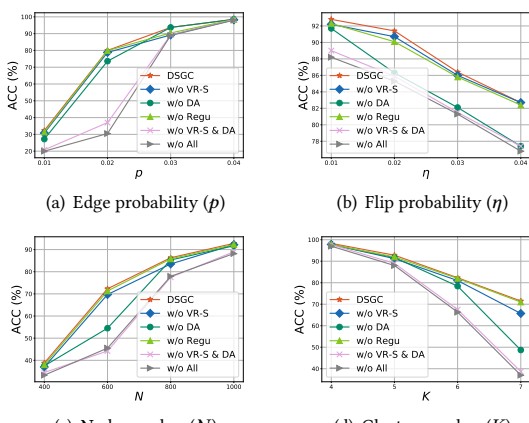

(a) Edge probability ($p$)  (b) Flip probability ($\eta$)

(c) Node number ($N$)  (d) Cluster number ($K$)

**Figure 4: Ablation study. (a)~(d) The ACC(%) performance vs. edge probability ($p$), flip probability ($\eta$), node number $N$ and cluster number $K$.**

**w/o Regu** is DSGC without Regularization term; **w/o VS-R & DA** is DSGC without VS-R and DA; **w/o DA & Regu** is DSGC without DA and Regu; **w/o VS-R & Regu** is DSGC without VS-R and Regu; and **w/o All** is DSGC without VS-R, DA, and Regu.

From the results depicted in Fig. 4, it is evident that: (i) *Performance trends*: As the edge probability ($p$) and the number of nodes ($N$) increase, the accuracy (ACC) of DSGC and its variants consistently improves. Conversely, increases in the sign flip probability ($\eta$) and the number of clusters ($K$) lead to a decline in ACC across all models. (ii) *Component impact*: DSGC outperforms all variants on all labeled signed graphs, demonstrating the significant role each component plays in enhancing clustering performance. Specifically, DA emerges as the most influential component, affirming its effectiveness in reinforcing the graph structure and improving node representations by adding strategically placed new edges.

## 5.4 Analysis of Violation Sign-Refine

To investigate **RQ3**, we analyzed the impact of applying Violation Sign-Refine (VS-R) on the performance of spectral clustering methods. VS-R was first used to pre-process and denoise signed graphs to generate new graphs. Then we compared the performance of all spectral methods before and after applying VS-R. Signed graphs were generated by fixing $N = 1000$ and varying ($K$, $\eta$, $p$), including SSBM (1000, $K$, 0.01, 0.02) with $K \in \{5, 6\}$, SSBM (1000, 5, 0.01, 0.04), and SSBM (1000, 10, $p$, 0.02) with $p \in \{0.01, 0.02\}$.

Table 2 shows that VS-R significantly improves the clustering performance across all tested spectral methods w.r.t. ACC and NMI by generating cleaner graphs with better clustering structure. Specifically, the performance increments vary inversely with the strength of the baseline methods—stronger baselines show smaller gains, whereas weaker baselines benefit more substantially from the VS-R preprocessing. VS-R also consistently reduces the *violation ratio*, defined as the ratio of the number of violated edges to the number of non-violated edges, across various graph configurations.

In addition to numerical analysis, Fig. 14 in App. D provides visual evidence of the impact of VS-R. the embeddings of new

**Table 1: Performance comparison of graph clustering methods on SSBM graphs with ACC (%) and NMI (%). Bold values indicate the best results; underlined values indicate the runner-up.**

| SSBM | $(N=1000, K=5, p=0.01, \eta)$ | | | | | | | | | | $(N=1000, K=10, p, \eta=0.02)$ | | | | | | | | | |
|---|---|---|---|---|---|---|---|---|---|---|---|---|---|---|---|---|---|---|---|---|
| SSBM | $\eta=0$ | | $\eta=0.02$ | | $\eta=0.04$ | | $\eta=0.06$ | | $\eta=0.08$ | | $p=0.01$ | | $p=0.02$ | | $p=0.03$ | | $p=0.04$ | | $p=0.05$ | |
| Metrics | ACC | NMI | ACC | NMI | ACC | NMI | ACC | NMI | ACC | NMI | ACC | NMI | ACC | NMI | ACC | NMI | ACC | NMI | ACC | NMI |
| $A^*$ | 71.60 | 41.06 | 68.10 | 35.95 | 62.20 | 27.86 | 41.60 | 15.25 | 43.80 | 12.65 | 16.70 | 3.90 | 2.13 | 7.89 | 42.80 | 24.43 | 79.70 | 62.52 | 93.30 | 86.60 |
| $\bar{L}_{sns}$ | 21.20 | 0.96 | 21.20 | 0.85 | 21.10 | 1.23 | 20.50 | 0.83 | 20.60 | 1.05 | 12.20 | 1.86 | 14.30 | 2.91 | 18.30 | 6.34 | 18.80 | 7.55 | 34.20 | 23.08 |
| $\bar{L}_{dns}$ | 41.70 | 16.39 | 38.30 | 12.47 | 31.70 | 6.45 | 31.20 | 6.46 | 29.10 | 3.75 | 15.70 | 2.91 | 19.10 | 5.80 | 27.60 | 12.29 | 46.90 | 30.57 | 83.50 | 68.80 |
| $\bar{L}$ | 20.30 | 0.79 | 20.30 | 0.79 | 20.30 | 0.79 | 20.30 | 0.79 | 20.30 | 0.79 | 10.70 | 1.75 | 10.70 | 1.75 | 10.70 | 1.94 | 10.70 | 2.02 | 10.70 | 1.75 |
| $L_{sym}$ | 75.80 | 46.49 | 69.40 | 37.06 | 62.30 | 28.46 | 48.20 | 19.35 | 47.70 | 15.68 | 16.00 | 2.69 | _19.60_ | 6.47 | 40.00 | 21.55 | 78.60 | 61.12 | 93.70 | 86.03 |
| BNC | 41.00 | 14.76 | 39.50 | 12.66 | 35.70 | 7.28 | 27.90 | 5.34 | 27.50 | 3.84 | 15.10 | 2.31 | 19.30 | 5.77 | 23.80 | 12.22 | 49.00 | 32.98 | 83.60 | 69.19 |
| BRC | 20.30 | 0.79 | 20.30 | 0.79 | 20.40 | 0.78 | 20.30 | 0.79 | 20.30 | 0.79 | 11.10 | 1.95 | 10.70 | 1.82 | 12.10 | 3.20 | 13.80 | 4.73 | 10.70 | 1.75 |
| SPONGE | 86.40 | 65.49 | _81.40_ | 55.73 | _71.70_ | 41.85 | _55.00_ | _28.61_ | _49.90_ | _22.21_ | 18.40 | 6.41 | 2.86 | 15.08 | 62.70 | 41.71 | 90.50 | 80.30 | 97.70 | 94.59 |
| $\text{SPONGE}_{sym}$ | 88.60 | _77.89_ | 67.60 | _57.00_ | 63.20 | _46.35_ | 35.00 | 20.90 | 32.20 | 12.12 | _19.90_ | 11.34 | 19.50 | _20.53_ | _81.60_ | **78.10** | _95.90_ | _90.92_ | _98.90_ | _97.30_ |
| DAEGC | 32.20 | 5.12 | 32.70 | 6.75 | 31.40 | 5.07 | 31.20 | 4.37 | 29.10 | 2.90 | 14.60 | _14.60_ | 15.90 | 3.22 | 17.10 | 4.38 | 18.50 | 7.43 | 21.80 | 11.70 |
| DFCN | 34.70 | 6.56 | 32.60 | 4.82 | 30.30 | 3.45 | 28.80 | 3.00 | 28.50 | 3.65 | 14.90 | 2.37 | 14.30 | 2.17 | 15.70 | 2.93 | 16.80 | 3.74 | 16.20 | 3.47 |
| DCRN | 48.40 | 23.18 | 44.40 | 18.80 | 43.70 | 21.07 | 37.10 | 11.40 | 33.30 | 10.37 | 16.70 | 3.61 | _19.60_ | 8.18 | 25.30 | 12.92 | 33.50 | 25.56 | 48.30 | 38.44 |
| Dink-net | 27.20 | 2.21 | 27.00 | 1.73 | 27.70 | 1.87 | 26.50 | 1.59 | 26.40 | 2.03 | 14.60 | 1.80 | 15.10 | 2.00 | 15.20 | 25.20 | 16.90 | 3.47 | 18.80 | 4.57 |
| DGCLUSTER | 20.30 | 0.79 | 20.30 | 0.79 | 20.30 | 0.79 | 20.30 | 0.79 | 20.30 | 0.79 | 10.40 | 1.75 | 10.60 | 1.75 | 10.70 | 1.75 | 10.70 | 1.75 | 10.60 | 1.75 |
| MAGI | 41.80 | 10.09 | 34.10 | 7.28 | 30.70 | 5.78 | 32.50 | 5.09 | 29.5 | 4.23 | 16.40 | 3.43 | 16.20 | 3.46 | 19.60 | 6.56 | 20.40 | 8.99 | 29.20 | 14.66 |
| DSGC | **95.30** | **85.40** | **90.80** | **73.60** | **82.80** | **57.30** | **66.50** | **33.50** | **57.70** | **23.30** | **28.70** | **13.10** | **64.90** | **44.80** | **85.40** | _71.90_ | **96.90** | **92.80** | **99.20** | **98.10** |
| SSBM | $(N, K=5, p=0.01, \eta=0)$ | | | | | | | | | | $(N=1000, K, p=0.01, \eta=0.02)$ | | | | | | | | | |
| SSBM | $N=300$ | | $N=500$ | | $N=800$ | | $N=1000$ | | $N=1200$ | | $K=4$ | | $K=5$ | | $K=6$ | | $K=7$ | | $K=8$ | |
| Metrics | ACC | NMI | ACC | NMI | ACC | NMI | ACC | NMI | ACC | NMI | ACC | NMI | ACC | NMI | ACC | NMI | ACC | NMI | ACC | NMI |
| A | 28.67 | 4.62 | 35.80 | 11.40 | 50.62 | 20.98 | 71.60 | 41.06 | 83.08 | 58.26 | 90.70 | 71.11 | 68.10 | 35.95 | 32.80 | 9.20 | 24.70 | 6.62 | 20.70 | 3.74 |
| $\bar{L}_{sns}$ | 21.67 | 2.69 | 21.60 | 1.78 | 21.25 | 1.44 | 21.20 | 0.96 | 21.67 | 1.48 | 25.30 | 0.70 | 21.20 | 0.85 | 16.40 | 1.36 | 16.00 | 2.09 |  |  |
| $\bar{L}_{dns}$ | 21.33 | 3.00 | 23.60 | 2.50 | 24.38 | 2.16 | 41.70 | 16.39 | 50.42 | 18.22 | 60.90 | 33.87 | 38.30 | 12.47 | 24.20 | 3.56 | 18.60 | 2.17 | 19.30 | 2.75 |
| $\bar{L}$ | 25.20 | 0.59 | 20.30 | 0.79 | 17.10 | 0.98 | 15.00 | 1.55 | 12.90 | 1.37 | 21.33 | 2.17 | 21.20 | 0.94 | 20.37 | 0.98 | 20.30 | 0.79 | 20.25 | 0.66 |
| $L_{sym}$ | 21.00 | 3.15 | 30.00 | 7.47 | 50.12 | 20.91 | 75.80 | 46.49 | 83.92 | 59.47 | 91.80 | 73.31 | 69.40 | 37.06 | 32.00 | 8.85 | 26.80 | 7.99 | 20.80 | 4.55 |
| BNC | 21.67 | 2.11 | 25.20 | 3.08 | 24.88 | 1.89 | 41.00 | 14.76 | 54.50 | 21.56 | 60.80 | 31.74 | 39.50 | 12.66 | 25.80 | 4.61 | 18.50 | 1.74 | 19.10 | 3.09 |
| BRC | 21.00 | 2.54 | 20.60 | 1.45 | 24.88 | 1.89 | 20.30 | 0.79 | 20.25 | 0.66 | 25.40 | 0.71 | 20.30 | 0.79 | 17.10 | 1.02 | 14.80 | 1.17 | 13.00 | 1.37 |
| SPONGE | 21.33 | 3.07 | 29.20 | 11.43 | 20.50 | 0.98 | 86.40 | 65.49 | _94.75_ | _83.37_ | _95.70_ | _84.24_ | _81.40_ | 55.73 | 43.10 | 20.83 | _45.30_ | 19.93 | 23.10 | 8.97 |
| $\text{SPONGE}_{sym}$ | 27.67 | _7.24_ | 35.20 | _15.56_ | _82.12_ | _66.43_ | _88.60_ | _77.89_ | 91.33 | 81.92 | 94.70 | 82.10 | 67.60 | _57.00_ | _62.90_ | _44.58_ | 32.70 | _21.65_ | _25.00_ | _10.52_ |
| DAEGC | 26.67 | 3.56 | 27.00 | 2.74 | 27.88 | 3.47 | 32.20 | 5.12 | 35.42 | 9.17 | 39.30 | 8.00 | 32.70 | 6.75 | 24.30 | 2.66 | 20.60 | 2.24 | 18.20 | 2.74 |
| DFCN | 28.33 | 4.95 | _36.80_ | 6.00 | 31.50 | 4.47 | 34.70 | 6.56 | 30.25 | 3.79 | 43.70 | 8.47 | 32.60 | 4.82 | 23.70 | 1.59 | 19.40 | 1.60 | 17.40 | 1.95 |
| DCRN | 28.00 | 4.53 | 33.40 | 8.69 | 32.00 | 10.69 | 48.40 | 23.18 | 49.83 | 29.91 | 67.30 | 39.13 | 44.40 | 18.80 | 33.60 | 12.89 | 24.20 | 6.66 | 19.80 | 4.47 |
| Dink-net | 29.33 | 2.74 | 28.60 | 2.41 | 27.38 | 1.75 | 27.20 | 2.21 | 29.33 | 2.46 | 35.00 | 2.76 | 27.00 | 1.73 | 21.90 | 1.61 | 20.70 | 1.55 | 18.00 | 1.74 |
| DGCLUSTER | 20.67 | 2.55 | 20.80 | 1.55 | 20.37 | 0.98 | 20.30 | 0.79 | 20.25 | 0.66 | 25.20 | 0.59 | 20.30 | 0.79 | 17.00 | 0.98 | 14.60 | 1.18 | 12.80 | 1.37 |
| MAGI | _33.00_ | 6.11 | 35.00 | 5.46 | 32.38 | 5.79 | 41.80 | 10.09 | 45.42 | 14.34 | 46.70 | 13.49 | 34.10 | 7.28 | 24.00 | 2.76 | 20.60 | 2.64 | 17.30 | 2.16 |
| DSGC | **37.70** | **10.30** | **54.60** | **30.80** | **89.40** | **71.40** | **95.30** | **85.40** | **98.40** | **94.40** | **97.40** | **90.10** | **90.80** | **73.60** | **70.90** | **45.10** | **51.30** | **23.90** | **35.90** | **14.40** |

**Table 2: Improvements in ACC (%) (↑) and NMI (%) (↑) and reductions in violation ratio (↓) with VS-R.**

| SSBM | $(1000, K, 0.01, 0.02)$ | | | | $(1000, 5, 0.01, \eta)$ | | $(1000, 10, p, 0.02)$ | | | |
|---|---|---|---|---|---|---|---|---|---|---|
| SSBM | $K=5$ | | $K=6$ | | $\eta=0.04$ | | $p=0.01$ | | $p=0.02$ | |
| Metrics | ACC | NMI | ACC | NMI | ACC | NMI | ACC | NMI | ACC | NMI |
| A | -12.10 | 1.49 | 15.00 | 15.89 | 13.70 | 0.74 | 9.00 | 3.42 | 19.80 | 27.24 |
| $\bar{L}_{sns}$ | 61.11 | 55.75 | 39.10 | 25.88 | 56.10 | 47.02 | 5.10 | 2.54 | 51.10 | 40.57 |
| $\bar{L}_{dns}$ | 44.20 | 44.28 | 33.10 | 24.69 | 45.50 | 41.77 | 2.50 | 1.90 | 46.10 | 38.18 |
| $\bar{L}$ | 0.10 | -0.01 | -0.1 | 0.00 | 0.10 | -0.01 | 0.00 | 0.00 | 0.00 | 0.00 |
| $L_{sym}$ | 12.80 | 19.74 | 25.10 | 19.78 | 16.40 | 22.15 | 3.80 | 3.84 | 45.70 | 36.68 |
| BNC | 42.70 | 43.87 | 32.10 | 24.72 | 41.70 | 41.18 | 2.00 | 26.80 | 47.20 | 39.31 |
| BRC | 0.1 | -0.01 | -0.1 | -0.04 | 0.00 | 0.00 | -0.7 | -0.2 | -0.3 | -0.06 |
| SPONGE | 2.70 | 4.35 | 14.70 | 8.30 | 5.60 | 6.38 | 0.30 | 0.50 | 29.40 | 24.15 |
| $\text{SPONGE}_{sym}$ | 17.10 | 9.34 | 9.70 | 8.99 | 12.40 | 3.64 | 5.10 | 3.71 | 25.40 | 22.50 |
| violation ratio | 2.72 | | 1.18 | | 3.48 | | 0.46 | | 0.3 | |

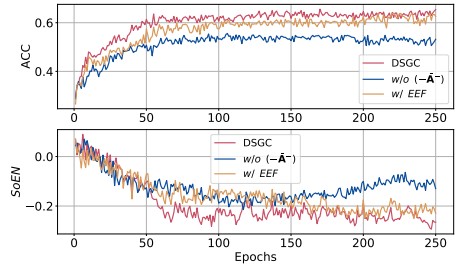

**Figure 5: The impact of the term $(-\bar{A}^-)$ and "EEF" principle on ACC (%) (Top) and *SoEN* (Bottom).**

graphs, displayed in the bottom row, exhibit clearer clustering boundaries than those of the original graphs in the top row.

## 5.5 Impact of Signed Encoder to Clustering

To address **RQ4**, we developed two variants of DSGC encoder, including DSGC w/o $(-\bar{A}^-)$ that replaces $(-\bar{A}^-)$ with $(\bar{A}^-)$, and DSGC w/ *EEF* that incorporates the "*the enemy of my enemy is my friend (EEF)*" principle from Balance Theory to DSGC. Both variants and DSGC used the layer number $L = 2$.

The positive and negative representations of DSGC w/ *EEF* are $Z^+_{eef} = Z^+ + (\bar{A}^-)^2 Z^{+(0)}$; $Z^-_{eef} = Z^-$ where $Z^+$ and $Z^-$ are the positive and negative representations computed by Eq. 8 in DSGC. Similarly, the positive and negative embeddings of DSGC *w/o* $(-\bar{A}^-)$ are $Z^+_{-a} = Z^+$; $Z^-_{-a} = -Z^-$.. We define a metric, *SoEN*, to measure the distance between nodes linked by negative edges:

$$SoEN = \frac{|\mathcal{E}^+|}{|\mathcal{E}^-|} \cdot \frac{\sum_{e_{ij}\in\mathcal{E}^-} s(z_i, z_j)}{\sum_{e_{i'j'}\in\mathcal{E}^+} s(z_{i'}, z_{j'})},$$

where $s(\cdot, \cdot)$ is the inner product, indicating the similarity between two nodes. Ideally, *SoEN* is a negative value and a lower *SoEN* indicates a greater distance between nodes connected by negative edges and a clearer clustering boundary. Fig. 5 illustrates the ACC and *SoEN* of DSGC and its variants. The results show that: (i) DSGC consistently outperforms its variants. Incorporating the "*EEF*" principle or altering the sign of $(-\bar{\mathbf{A}}^-)$ significantly impacts clustering performance because DSGC achieves lower *SoEN* along with epochs than its variants. This demonstrates its advantage in separating nodes linked by negative edges, leading to clearer clustering boundaries and larger inter-cluster variances. (ii) The term $(-\bar{\mathbf{A}}^-)$ has higher impact than the inclusion of *EEF*, suggesting the original negative edge handling in DSGC is critical for maintaining clear cluster separations.

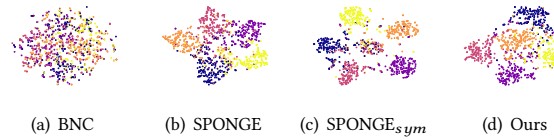

(a) BNC     (b) SPONGE     (c) SPONGE$_{sym}$     (d) Ours

**Figure 6: Visualization of clustering results from different algorithms. The ground truth class number is** $5$.

## 5.6 Visualization

We utilized t-SNE to visualize the embeddings produced by DSGC and several strong baselines, including BNC [5], BRC [5], SPONGE [6], and SPONGE$_{sym}$ [6], on SSBM ($N = 1000$, $K = 5$, $p = 0.01$, $\eta = 0.02$) in Fig. 6. Both BNC and BRC exhibit mode collapse, where most nodes are grouped into one or a few clusters. SPONGE and SPONGE$_{sym}$ show improved clustering structures. However, SPONGE lacks a clear boundary between clusters while SPONGE$_{sym}$ appears to form 6 clusters with a central cluster where nodes from different true clusters are mixed. This indicates its potential issue with handling nodes connected by negative edges, which are typically located at cluster boundaries. In contrast, DSGC successfully pushes nodes linked by negative edges apart, effectively eliminating the central cluster phenomenon in SPONGE$_{sym}$. This result is attributed to the exclusion of the "*EEF*" principle and the incorporation of the term $(-\mathbf{A}^-)$ in the graph encoder.

## 5.7 Unlabeled Graphs

We also evaluated DSGC on unlabeled real-world signed graphs, S&P1500 [6] and Rainfall [6], comparing it against three baselines, BRC [5], BNC [5], and SPONGE_sym [6]. The adjacency matrices of these graphs were sorted by predicted cluster membership to visually assess clustering outcomes.

**Rainfall.** Following [6], we analyzed the clustering structures for $K = \{5, 10\}$ in Fig. 7, where blue and red denote positive and negative edges, respectively[2]. Both BRC and BNC fail to identify the expected number of clusters, resulting in model collapse. In contrast, DSGC successfully identifies the specified clusters (5 or 10)

---

[2]Darker blue diagonal blocks indicate more cohesive clusters, while darker pink non-diagonal blocks signify stronger negative relationships between clusters, enhancing the clarity of the clustering semantics.

and exhibits higher ratios of positive internal edges and stronger negative inter-cluster edges compared to SPONGE, indicating more cohesive and well-defined clusters.

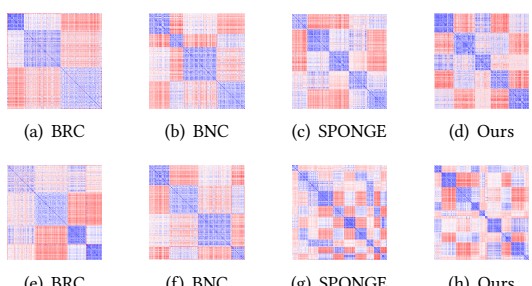

(a) BRC    (b) BNC    (c) SPONGE    (d) Ours

(e) BRC    (f) BNC    (g) SPONGE    (h) Ours

**Figure 7: Sorted adjacency matrix for the Rainfall dataset with** $K = 5$ **(top row) and** $K = 10$ **(bottom row).**

**S&P1500.** Fig. 8 shows the clustering structures for $K = \{5, 10\}$. BRC and BNC suffer model collapse, placing most nodes into a single large, sparse cluster. In contrast, DSGC produces clear, compact clusters with significantly higher ratios of positive to negative internal edges than the entire graph, indicating more effective clustering that even surpasses SPONGE in identifying relevant groupings.

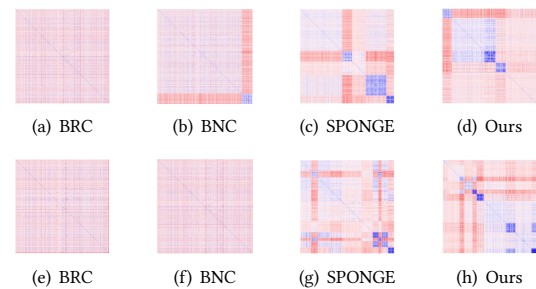

(a) BRC    (b) BNC    (c) SPONGE    (d) Ours

(e) BRC    (f) BNC    (g) SPONGE    (h) Ours

**Figure 8: Sorted adjacency matrix for S&P**$1500$ **with** $K = 5$ **(top row) and** $K = 10$ **(bottom row).**

## 6 CONCLUSION

In this paper, we introduce DSGC, a novel deep signed graph clustering method, to enhance the clarity of cluster boundaries by effectively utilizing positive and negative edge connections for node partitioning. Existing approaches generally rely on the Social Balance Theory, which is primarily suitable for 2-way clustering. In contrast, DSGC leverages the Weak Balance Theory to address more general $K$-way clustering without the need for explicit labels. DSGC first introduces two pre-processing techniques, VS-R and DA, to denoise and structurally enhance signed graphs before clustering. Then, DSGC constructs a clustering-oriented signed neural network that produces more discriminative representations, specifically for nodes linked negatively. By optimizing a non-linear transformation for node clustering assignments, DSGC significantly outperforms existing methods, establishing clearer and more meaningful cluster distinctions in complex multi-cluster scenarios.

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

## A   Sociological Theories

The study of clusterability in signed graphs can be traced back to the foundational *Social Balance Theory* [14], stating that a signed undirected network without attributes can naturally exhibit a global structure conducive to 2-way clustering.

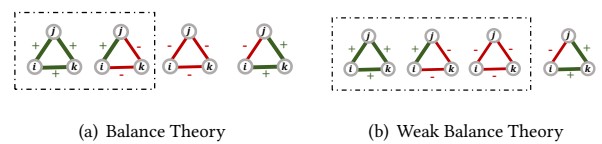

(a) Balance Theory          (b) Weak Balance Theory

**Figure 9: Illustrations of sociological theories. Triads in boxes are considered as balanced and weakly balanced, respectively.**

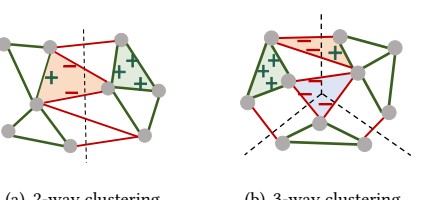

(a) 2-way clustering          (b) 3-way clustering

**Figure 10: Illustrative comparison of Balance and Weak Balance Theory in signed graph clustering. (a) 2-way clustering. The orange triangle proves the principle "*the enemy of my enemy is my friend*". (b) $K$-way ($K = 3$) clustering. The blue triangle proves the principle "*the enemy of my enemy might be my enemy*".**

THEOREM 1 (**SOCIAL BALANCE THEORY**).   *The signed network is balanced if and only if (i) all the edges are positive, or (ii) the node set can be partitioned into two mutually exclusive subsets, such that all edges within the same subset are positive and all edges between the two subsets are negative.*

Social Balance Theory induces four fundamental principles: "*the friend of my friend is my friend (FFF)*", "*the enemy of my friend is my enemy (EFE)*", "*the friend of my enemy is my enemy (FEE)*", and "*the enemy of my enemy is my friend (EEF)*". A signed network is balanced if it does not violate these principles. For example, triads with an even number of negative edges are balanced, as shown by the first two triads in Fig. 9(a), which have 0 and 2 negative edges, respectively. These principles are traditionally applied to 2-way clustering.

To accommodate $K$-way clustering, [7] proposes Weak Balance Theory, a relaxed version of Social Balance Theory.

THEOREM 2 (**WEAK BALANCE THEORY**).   *The signed network is weakly balanced if and only if (i) all the edges are positive, or (ii) all nodes can be partitioned into $K \in \mathbb{N}^+$ disjoint sets, such that positive edges exist only within clusters, and negative edges exist only between clusters.*

Conceptually, Weak Balance Theory replaces the "*EEF*" principle in Social Balance Theory with "*the enemy of my enemy might be my enemy (EEE)*". Accordingly, the first three triads in Fig. 9(b) are considered weakly balanced. Importantly, the "*EEE*" principle, applicable for $K$-way ($K > 2$) clustering, allows nodes in a triangle to belong to three different clusters (e.g., the blue triangle in Fig. 2(b)), illustrating a relaxation of the stricter Social Balance Theory.

**Comparison of Social Balance Theory and Weak Balance Theory.** The partition $\{C_1, \ldots, C_K\}$ of a signed graph $\mathcal{G}$ satisfying both theories can be uniformly defined such that the following conditions hold:

$$\begin{cases} \mathbf{A}_{ij} > 0 & (e_{ij} \in \mathcal{E}) \cap (v_i \in C_k) \cap (v_j \in C_k) \\ \mathbf{A}_{ij} < 0 & (e_{ij} \in \mathcal{E}) \cap (v_i \in C_k) \cap (v_j \in C_l)(k \neq l) \end{cases} \quad (17)$$

where $\mathbf{A}_{ij}$ is the weight of edge $e_{ij}$ and $0 < k, l < K$. However, the "*EEE*" principle is specific to $K$-clusterable ($K > 2$) networks (e.g., Fig. 10(b)) and does not appear in 2-clusterable systems (Fig. 10(a)).

Recent literature [8, 16, 17, 24, 24, 38, 43] has primarily leveraged Social Balance Theory principles to improve node representations for signed graphs, potentially overlooking the broader applicability of Weak Balance Theory in datasets with more than 2 antagonistic groups, especially when explicit labels are lacking. Our work aims to fully explore Weak Balance Theory and its principles in the design of a signed graph encoder for $K$-way clustering.

## B Analyzing the impact of the signed encoder to node representations and clustering boundary

In this section, we specifically analyze the principles of Weak Balance Theory implied in positive and negative aggregation functions (Eq. (9) and Eq. (10)) and the term $(-\bar{\mathbf{A}}^-)$ in Eq. (10) for the perspective of their impact to the node representations and the clustering boundaries.

**The term $(-\bar{\mathbf{A}}^-)$.** The minus sign "$-$" helps push nodes linked by negative edges further apart in the latent space. For example, in Fig. 11 (a), the node $u$ has three "*friend neighbors*", $v_1$, $v_2$, and $v_3$. The positive embedding $u^+$ of $u$ is placed at the mean of these three "*friend neighbors*" according to Eq. (9), thus narrowing the distance between them and its central node $u$. In Fig. 11 (b), the node $u$ has three "*enemy neighbors*", $v_4$, $v_5$, and $v_6$. "$-$" in the term $(-\bar{\mathbf{A}}^-)$ indicates that two vertices with a negative edge should be placed on opposite sides. Then, the negative embedding $u^-$ is placed at the mean of $-v_4$, $-v_5$, and $-v_6$, which are the opposite coordinates of $v_4$, $v_5$, and $v_6$, respectively. This leads to further distance between $u$ and its "*enemy neighbors*". As nodes linked negatively are likely to be located at the cluster boundary, pushing them away from each other will create clearer cluster boundaries, thus effectively increasing inter-cluster variances. Section 5.5 quantitatively analyzes the effect of $(-\bar{\mathbf{A}}^-)$ on node embeddings.

**Positive and Negative aggregation.** Eq. (9) aggregates the node embeddings of all $l$-hop "*friend neighbors*" along the $l$-length positive walk (Dfn 1), implying the principle "*the friend of my friend is my friend (FFF)*" and its transitivity. It pulls "*friend neighbors*" within $L$-hop toward the central node, thus reducing intra-cluster variances. Eq. (10) aggregates the node embeddings of all $l$-hop "*enemy neighbors*" along the $l$-length negative walk (Dfn 1), implying the principles of "*the enemy of my friend is my enemy (EFE)*", "*the friend of my enemy is my enemy (FEE)*", and the transitivity of "*FFF*". Importantly, we no longer consider the specific principle "*the Enemy of my Enemy is my Friend (EEF)*" of Social Balance so that the distance between nodes linked negatively can effectively increase, which can be verified by quantitatively comparing our DSGC and its variant that incorporates "*EEF*". Taking a signed graph with

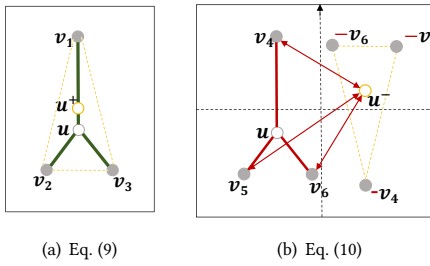

(a) Eq. (9)  (b) Eq. (10)

**Figure 11: Impact of the term $(-\bar{\mathbf{A}}^-)$ in Eq. (10) when both $\tau^+$ and $\tau^-$ are 0. (a) The positive embedding $u^+$ is placed at the mean of its "*friend neighbors*", including $v_1$, $v_2$, and $v_3$. (b) Due to "-" in the term, the negative embedding $u^-$ is placed at the mean of its "*enemy neighbors*'" antipodal points, including $-v_4$, $-v_5$, and $-v_6$, resulting in $u^-$ further away from $v_4$, $v_5$, and $v_6$.**

$L = 2$ as an example, Fig. 12 illustrates our positive and negative aggregation rules in Eq. (9) and (10).

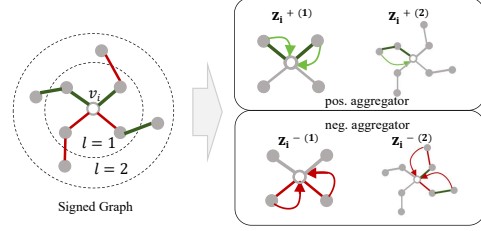

**Figure 12: The illustration of positive and negative aggregations in Eq. (9) and Eq. (10) on a signed graph with the central node $v_i$ and its 2-hop neighbors.**

## C Hyperparameter sensitivity analysis

This section explores the sensitivity of DSGC's performance to variations in its hyperparameters, specifically focusing on $\delta^+$, $\delta^-$, $m^+$, and $m^-$. $\delta^+$, $\delta^-$ are thresholds that determine the confidence levels for nodes being classified as *effective friends* or *effective enemies*, respectively. $m^+$ and $m^-$ control the augmentation of positive and negative edge densities within and across clusters, respectively. We used synthetic signed graphs from SSBM (1000, 5, 0.01, 0.02) for this analysis. The results illustrated in Fig. 13 show that: (i) Optimal performance is achieved when both $\delta^+$ and $\delta^-$ are set to 1. Increasing $\delta^+$ generally worsens accuracy (ACC) as less noisy edges are effectively refined. (ii) Both excessively high and low values of $m^+$ degrade clustering performance due to imbalances in capturing local versus more extended neighborhood information. Setting $m^+$ to 3 and $m^-$ to 2 achieve optimal performance.

**Table 3: Performance comparison of signed graph clustering on SSBM with ARI (%) and F1 (%). Bold values indicate the best results; underlined values indicate the runner-up.**

| SSBM | \multicolumn (N=1000, K=5, p=0.01, $\eta$) | | | | | | | | | | (N=1000, K=10, p, $\eta$ = 0.02) | | | | | | | | | |
|---|---|---|---|---|---|---|---|---|---|---|---|---|---|---|---|---|---|---|---|---|
| SSBM | $\eta = 0$ | | $\eta = 0.02$ | | $\eta = 0.04$ | | $\eta = 0.06$ | | $\eta = 0.08$ | | p=0.01 | | p=0.02 | | p=0.03 | | p=0.04 | | p=0.05 | |
| Metrics | ARI | F1 | ARI | F1 | ARI | F1 | ARI | F1 | ARI | F1 | ARI | F1 | ARI | F1 | ARI | F1 | ARI | F1 | ARI | F1 |
| A | 41.66 | 71.58 | 35.90 | 68.26 | 28.09 | 62.14 | 12.03 | 41.74 | 9.94 | 43.98 | 0.77 | 16.13 | 3.01 | 21.32 | 16.85 | 43.04 | 60.21 | 79.80 | 86.91 | 93.91 |
| $\bar{L}_{sns}$ | 0.01 | 10.17 | 0.01 | 10.36 | 0.01 | 9.01 | 0.00 | 8.36 | 0.00 | 8.08 | 0.00 | 8.17 | 0.11 | 11.34 | 1.68 | 13.48 | 3.40 | 11.20 | 15.78 | 25.01 |
| $\bar{L}_{dns}$ | 15.11 | 31.92 | 11.42 | 30.52 | 5.32 | 24.17 | 6.11 | 24.47 | 3.05 | 21.12 | 0.62 | 12.63 | 2.14 | 18.09 | 6.26 | 27.56 | 23.36 | 45.17 | 67.10 | 83.40 |
| $\bar{L}$ | 0.00 | 7.29 | 0.00 | 7.29 | 0.00 | 7.29 | 0.00 | 7.29 | 0.00 | 7.29 | 0.00 | 3.22 | 0.00 | 3.22 | 0.00 | 3.22 | 0.00 | 3.21 | 0.00 | 3.22 |
| $L_{sym}$ | 48.56 | 75.91 | 38.62 | 69.37 | 28.83 | 62.26 | 16.16 | 48.49 | 14.01 | 47.88 | 0.49 | 15.91 | 2.41 | 19.64 | 14.66 | 39.91 | 58.49 | 78.69 | 86.48 | 93.69 |
| BNC | 14.55 | 35.93 | 12.32 | 32.50 | 6.80 | 29.86 | 4.56 | 23.78 | 1.67 | 22.18 | 0.17 | 13.06 | 2.31 | 18.78 | 6.09 | 23.47 | 24.89 | 48.76 | 67.28 | 83.51 |
| BRC | 0.00 | 7.29 | 0.00 | 7.29 | 0.00 | 7.48 | 0.00 | 7.29 | 0.00 | 7.29 | 0.00 | 3.88 | 0.00 | 3.22 | 0.04 | 5.56 | 0.50 | 6.40 | 0.00 | 3.22 |
| SPONGE | 68.90 | 86.39 | 58.65 | 81.42 | 41.36 | 71.73 | 24.28 | 50.20 | 17.22 | 45.82 | 1.61 | 15.17 | 8.25 | 28.01 | 35.52 | 6.31 | 80.11 | 90.48 | 94.94 | 97.69 |
| $SPONGE_{sym}$ | 71.34 | 89.38 | 46.42 | 63.15 | 38.31 | 58.64 | 14.79 | 21.07 | 9.02 | 18.91 | 1.64 | 17.44 | 7.94 | 9.90 | 67.53 | 79.43 | 90.97 | 95.93 | 97.56 | 98.90 |
| DAEGC | 3.66 | 27.98 | 5.27 | 26.81 | 3.65 | 26.57 | 3.09 | 26.55 | 2.03 | 24.29 | 0.26 | 11.99 | 0.73 | 12.78 | 1.05 | 15.40 | 2.51 | 16.14 | 4.93 | 19.97 |
| DFCN | 4.26 | 35.07 | 3.10 | 32.73 | 2.09 | 30.42 | 1.71 | 27.97 | 1.92 | 28.51 | 0.23 | 14.74 | 0.19 | 13.85 | 0.39 | 14.39 | 0.83 | 16.48 | 0.68 | 15.83 |
| DCRN | 11.20 | 48.95 | 9.10 | 44.13 | 11.34 | 40.30 | 4.05 | 36.09 | 4.05 | 34.72 | 0.84 | 13.95 | 2.59 | 18.99 | 5.45 | 25.27 | 13.70 | 32.84 | 24.72 | 49.26 |
| Dink-net | 1.37 | 26.40 | 0.93 | 26.12 | 1.09 | 27.03 | 0.90 | 26.43 | 1.15 | 26.22 | 0.00 | 14.37 | 0.06 | 15.08 | 0.22 | 13.80 | 0.83 | 15.85 | 1.54 | 15.56 |
| DGCLUSTER | 0.00 | 7.29 | 0.00 | 7.29 | 0.00 | 7.29 | 0.00 | 7.29 | 0.00 | 7.29 | 0.00 | 2.63 | 0.00 | 3.22 | 0.00 | 3.22 | 0.00 | 3.22 | 0.00 | 3.02 |
| MAGI | 9.47 | 39.24 | 6.69 | 28.48 | 4.42 | 28.25 | 4.03 | 30.40 | 3.09 | 26.75 | 0.79 | 13.83 | 1.03 | 14.45 | 2.59 | 19.62 | 4.06 | 20.18 | 7.98 | 29.16 |
| DSGC | **88.50** | **94.30** | **78.30** | **90.80** | **61.60** | **82.80** | **34.40** | **66.40** | **23.50** | **57.60** | **6.90** | **29.10** | **38.40** | **65.30** | **70.40** | **85.30** | **93.20** | **96.90** | **98.40** | **99.40** |

| SSBM | \multicolumn (N, K = 5, p = 0.01, $\eta$ = 0) | | | | | | | | | | (N = 1000, K, p = 0.01, $\eta$ = 0.02) | | | | | | | | | |
|---|---|---|---|---|---|---|---|---|---|---|---|---|---|---|---|---|---|---|---|---|
| SSBM | N = 300 | | N = 500 | | N = 800 | | N = 1000 | | N = 1200 | | K=4 | | K=5 | | K=6 | | K=7 | | K=8 | |
| Metrics | ARI | F1 | ARI | F1 | ARI | F1 | ARI | F1 | ARI | F1 | ARI | F1 | ARI | F1 | ARI | F1 | ARI | F1 | ARI | F1 |
| A | 0.64 | 25.13 | 6.24 | 35.70 | 15.11 | 51.42 | 41.66 | 71.58 | 62.24 | 83.10 | 76.81 | 90.69 | 35.90 | 68.26 | 6.44 | 31.68 | 3.45 | 24.20 | 1.48 | 19.77 |
| $\bar{L}_{sns}$ | 0.00 | 11.42 | 0.01 | 11.00 | 0.02 | 9.15 | 0.01 | 10.17 | 0.04 | 10.54 | 0.00 | 10.81 | 0.01 | 10.36 | 0.00 | 7.98 | 0.01 | 9.90 | 0.07 | 11.16 |
| $\bar{L}_{dns}$ | 0.01 | 9.36 | 0.27 | 17.36 | 0.46 | 15.22 | 15.11 | 31.92 | 16.88 | 49.98 | 32.63 | 53.01 | 11.42 | 30.52 | 2.10 | 18.55 | 0.77 | 15.11 | 0.96 | 16.26 |
| $\bar{L}$ | 0.00 | 9.95 | 0.00 | 12.35 | 0.00 | 7.44 | 0.00 | 7.29 | 0.00 | 7.18 | 0.00 | 10.42 | 0.00 | 7.29 | 0.00 | 5.59 | 0.00 | 4.98 | 0.00 | 3.59 |
| $L_{sym}$ | 0.02 | 8.71 | 4.86 | 29.81 | 19.16 | 50.35 | 48.56 | 75.91 | 63.80 | 83.91 | 79.31 | 91.79 | 38.62 | 69.37 | 6.17 | 31.87 | 4.93 | 26.51 | 1.96 | 20.80 |
| BNC | 0.00 | 11.06 | 0.62 | 18.09 | 0.60 | 15.88 | 14.55 | 35.93 | 21.13 | 53.83 | 30.89 | 53.48 | 12.32 | 32.50 | 3.51 | 20.71 | 0.65 | 15.29 | 1.34 | 16.76 |
| BRC | 0.00 | 8.71 | 0.00 | 8.92 | 0.00 | 7.69 | 0.00 | 7.29 | 0.00 | 7.18 | 0.00 | 10.83 | 0.00 | 7.29 | 0.00 | 5.76 | 0.00 | 4.59 | 0.00 | 3.79 |
| SPONGE | 0.01 | 9.38 | 3.72 | 20.06 | 25.22 | 58.50 | 68.90 | 86.39 | 87.28 | 94.75 | 88.84 | 95.71 | 58.65 | 81.42 | 16.74 | 35.71 | 15.30 | 42.37 | 3.86 | 19.64 |
| $SPONGE_{sym}$ | 0.85 | 22.87 | 4.65 | 30.32 | 58.48 | 83.23 | 71.34 | 89.38 | 78.18 | 91.82 | 86.32 | 94.70 | 46.42 | 63.15 | 31.01 | 65.55 | 9.08 | 30.05 | 3.87 | 19.21 |
| DAEGC | 0.63 | 21.81 | 0.68 | 22.89 | 1.72 | 22.66 | 3.66 | 27.98 | 6.74 | 29.28 | 6.47 | 36.34 | 5.27 | 26.81 | 1.27 | 19.82 | 0.85 | 15.16 | 0.80 | 15.89 |
| DFCN | 1.23 | 26.14 | 3.94 | 36.00 | 2.75 | 31.70 | 4.14 | 34.61 | 2.34 | 30.45 | 6.61 | 44.18 | 3.10 | 32.73 | 0.72 | 23.27 | 0.40 | 19.14 | 0.47 | 17.23 |
| DCRN | 0.67 | 26.21 | 3.59 | 33.40 | 3.50 | 30.88 | 11.20 | 48.95 | 18.67 | 50.84 | 29.98 | 68.75 | 9.10 | 44.13 | 6.09 | 31.83 | 2.61 | 24.23 | 1.10 | 17.04 |
| Dink-net | 0.70 | 28.18 | 1.33 | 25.38 | 0.97 | 27.12 | 1.37 | 26.40 | 1.72 | 29.36 | 2.68 | 32.11 | 0.93 | 26.12 | 0.60 | 21.79 | 0.40 | 18.95 | 0.28 | 16.55 |
| DGCLUSTER | 0.01 | 8.05 | 0.00 | 8.30 | 0.00 | 7.44 | | | 0.00 | 7.18 | 0.00 | 10.42 | 0.00 | 7.29 | 0.00 | 5.39 | 0.00 | 4.19 | 0.00 | 3.54 |
| MAGI | 3.99 | 31.39 | 3.74 | 35.09 | 4.55 | 30.30 | 9.32 | 38.42 | 13.19 | 42.20 | 13.64 | 42.45 | 6.74 | 27.46 | 1.52 | 22.54 | 1.49 | 16.92 | 0.46 | 15.46 |
| DSGC | **6.80** | **37.40** | **34.10** | **66.60** | **75.40** | **89.30** | **88.50** | **95.30** | **96.10** | **98.40** | **93.20** | **97.40** | **78.30** | **90.80** | **44.40** | **70.50** | **21.10** | **50.90** | **10.00** | **35.90** |

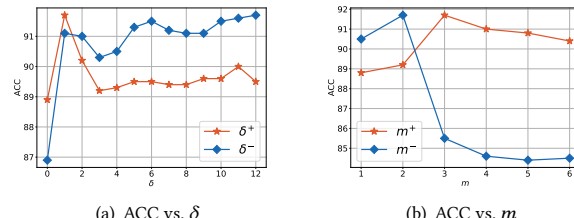

(a) ACC vs. $\delta$    (b) ACC vs. $m$

**Figure 13: Hyperparameter analysis on the signed graph.**

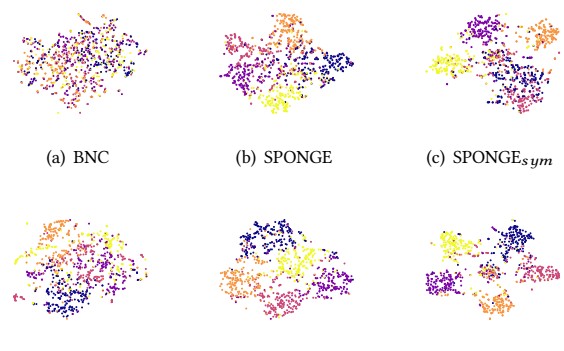

(a) BNC    (b) SPONGE    (c) $SPONGE_{sym}$

(d) BNC+VS-R    (e) SPONGE+VS-R    (f) $SPONGE_{sym}$+VS-R

**Figure 14: Node embeddings of three signed spectral methods before and after applying VS-R. Different colors represent different clusters.**

# D  Visualization of signed spectral clustering after applying VS-R

In addition to numerical analysis, Fig. 14 provides visual evidence of the impact of VS-R. Utilizing $t$-SNE, we compared the embeddings of original and denoised graphs learned by strong spectral methods on SSBM (1000, 5, 0.01, 0.04). We can observe that the embeddings of new graphs, displayed in the bottom row, exhibit clearer clustering boundaries than those of the original graphs in the top row. That is, spectral methods, including BNC, SPONGE, and $SPONGE_{sym}$, achieve enhanced clustering performance on cleaner graph structures after employing VS-R.

## E  Implementation Setting

All experiments are implemented on PYtorch. The length of positive and negative walks $L'$ in VS-R is set to 3. The balance parameter $\lambda$ in loss $\mathcal{L}$ is set to 0.03. The layer number $L$ in our graph encoder is set to 2. Node features $\mathbf{X}$ are derived from the $K$-dimmensional embeddings corresponding to the largest $K$ eigenvalues of the symmetrized adjacency matrix. The hidden dimension $d$ is 32 in the our cluster-specific signed graph clustering. Besides, the hyperparameters sensitivity analysisi of $\delta^+$, $\delta^-$, $m^+$, and $m^-$ in Eq. (5) and Eq. (6) can be found in App. C. Following [22], we change a signed graph to an unsigned graph by revising all negative edges to positive edges, which are inputted to above unsigned graph clustering methods (DAEGC, DFCN, DCRN, Dink-net, DGCLUSTER MAGI).

## F  ARI and F1 score of overall performance

This results in ARI and F1 score for DSGC and all baselines are reported in Table 3. They are generally consistent with the ACC and NMI results presented in the main text, reinforcing the conclusions drawn from those analyses. We can observe: (i) *Superior performance:* DSGC still significantly outperforms all baseline models in ARI and F1 score. (ii) *Robustness:* DSGC exhibits notably superior performance on all 20 labeled signed graphs in ARI and F1 score. This observation highlights the effectiveness and robustness of our approach regarding $\eta$, $p$, $N$, and $K$. (iii) *Comparative analysis:* In terms of ARI and F1 score, while unsigned clustering methods (DAEGC, DFCN, and DCRN) generally outperform non-deep spectral methods due to their advanced representation learning capabilities, DSGC still maintains a significant advantage, confirming that the specialized design of DSGC is effective for the unique challenges of signed graph clustering.

Received 20 February 2007; revised 12 March 2009; accepted 5 June 2009

