# OpenReview forum: "Robust Deep Signed Graph Clustering via Weak Balance Theory"
_ACM.org/TheWebConf/2025/Conference — WWW 2025 Poster_

### Official Review · Reviewer_f1yf · 2024-11-30

**Novelty:** 5
**Technical Quality:** 5

**Review:**

In this paper the authors propose a Deep Signed Graph Clustering (DSGC) approach. Identifying drawbacks of traditional approaches to signed graph clustering, such as sensitivity to noise and overreliance on the principles of social balance theory, they propose two distinct mechanisms, “violation sign-refine” for sign correction and “density-based augmentation” to increase the density of positive edges within the clusters and negative between them. Then, a signed graph convolutional network is applied to learn node embeddings based on weak balance theory principle. In the final stage, clustering is performed through an optimization that finds partitioning with minimal violations. The approach is evaluated on a number of synthetic and real datasets, demonstrating solid performance compared to the alternatives.

The paper is interesting and relevant to the scope of the conference. In general, it is relatively well written, save for occasional typos and the lack of in-depth explanations for some of the introduced concepts.

For example, throughout the paper the concept of high-order (neighbor) information is mentioned several times without proper introduction. This should be better explained especially since in graph theory high-order relations are often attributed to e.g. hypergraphs where a hyperedge can connect arbitrary number of nodes, in contrast to pairwise edges in graphs.

In Figure 2 it is unclear which part of the caption corresponds to sub-figure (a) and which to (b), which makes the associated part of Introduction difficult to understand. Aside from the caption, the annotations in Figure 2 should be improved.

In Section 5.1.1 a number of synthetic and real-life datasets are briefly mentioned. If all of them are originally introduced in [6], it should be stated more explicitly.

In Section 5.1.3 evaluation measures are briefly introduced. While Adjusted Rand Index and Normalized Mutual Information are fairly standard, it is unclear why accuracy and F1 are used as well. Namely, in their standard form accuracy and F1 score are typically used for classification and, therefore, inapplicable (or at least poor choices) for evaluating partitionings. For example, it is easy to think of trivial partitioning examples that would be identical, but still yield 0 accuracy. In any case, this should be better explained and motivated.

The paper should be carefully proofread to remove occasional typos. Below are some examples:

- Figure.1: Filp-edge Ratio → Flip-edge Ratio?
- Page 2, Line 145: “we propose eep” → probably missing Deep?
- Section 5.1.1: Follow SPONGE → Following SPONGE.

**Questions:**

- What high-order neighbor information are the authors referring to?
- Which part of Figure 2 caption corresponds to sub-figure (a) and which to sub-figure (b)?
- What is the motivation for using accuracy and F1 in addition to more standard clustering evaluation measures ARI and NMI?

**Reviewer Confidence:**

3: The reviewer is confident but not certain that the evaluation is correct

**Scope:**

3: The work is somewhat relevant to the Web and to the track, and is of narrow interest to a sub-community

---

### Official Review · Reviewer_AgZJ · 2024-12-02

**Novelty:** 5
**Technical Quality:** 5

**Review:**

This paper presents DSGC, an innovative deep signed graph clustering method designed to improve the definition of cluster boundaries by effectively leveraging both positive and negative edge connections for node partitioning. While existing methods often rely on Social Balance Theory, which is primarily suited for 2-way clustering, DSGC adopts the Weak Balance Theory to enable more general
K-way clustering without requiring explicit labels.

The proposed method incorporates two pre-processing techniques, VS-R and DA, to denoise and enhance the structural properties of signed graphs before clustering. DSGC then employs a clustering-oriented signed neural network to generate more discriminative node representations, particularly for nodes connected by negative edges. By optimizing a non-linear transformation for node clustering assignments, DSGC delivers significantly better performance compared to existing approaches, producing clearer and more meaningful cluster separations in complex multi-cluster scenarios.

The specific contributions made in the paper are:

--The authors develop DSGC, the first Deep Signed Graph Clustering
framework, by leveraging Weak Balance Theory.
--They design two graph rewiring strategies to denoise and
augment the overall network topology
--They propose a task-oriented signed graph encoder to learn
more discriminative representations, particularly for nodes
connected by negative edges.

Strengths of the paper:
--Related work and motivation are both reasonably strong
--The notation and formalism is a bonus
--The framework, while complex, has substance to it and is being empirically validated.

That being said, I do believe that the notation can at times be overly cumbersome, and I'd like the authors to try and simplify it if possible.

Experimentally, the authors state the following research questions. RQ1: Can
DSGC achieve state-of-the-art clustering performance on signed
graphs without any labels? RQ2: How does each component con-
tribute to the effectiveness of DSGC? RQ3: How does the Violation
Sign-Refine (VS-R) impact signed topology structures by correcting
noisy edges? RQ4: How do the strategies in the signed encoder,
specifically abandoning the “EEF ” principle and the minus sign in
term (−¯A−), contribute to forming wider clustering boundaries?

Experimental performance is laid out quite substantively, and generally speaking, the
authors' claims on the RQs are borne out.

There are some minor proofreading issues that the authors should correct. For example, 'eep' on Page 2, etc.

**Questions:**

Can you comment a little more on the scalability of the method?

**Reviewer Confidence:**

3: The reviewer is confident but not certain that the evaluation is correct

**Scope:**

4: The work is relevant to the Web and to the track, and is of broad interest to the community

---

### Official Review · Reviewer_G9xJ · 2024-12-03

**Novelty:** 3
**Technical Quality:** 5

**Review:**

The authors proposed a deep clustering method for signed graphs using two ad-hoc preprocessing techniques inspired by weak balance theory: (1) Violation Sign-Refine (VS-R) and (2) Density-Based Augmentation. They claim that these techniques generate cleaner graphs with better clustering structures. The method is shown to achieve better performance on a synthetic model, and the authors also provide some results for three real-world datasets. However, the evaluations of real data are not convincing.

It is important to note that there is no ground truth theorem for community detection algorithms [Peel, L., Larremore, D. B., & Clauset, A. (2017). The ground truth about metadata and community detection in networks. Science Advances, 3(5), e1602548.]. Therefore, the most robust way to evaluate clustering methods is through indirect approaches like link prediction [Ghasemian, A., Hosseinmardi, H., & Clauset, A. (2019).].

Given this context, I am not fully convinced that the proposed ad-hoc preprocessing techniques perform as claimed. This is especially true considering the following issues. The paper could benefit from addressing the ambiguities and concerns outlined below.

**Questions:**

1. The authors defined \mu^{-}(i,j) as the walk of length "l" with exactly one negative edge. However, they did not explain the rationale behind this choice. To my understanding, this could be an approximation, as the original definition in [10], from which the authors derive their definition, seems more logical. The original definition considers all unbalanced walks (i.e., walks with an odd number of negative edges), while the authors retain only walks with exactly one negative edge as the unbalanced contribution of the definition.

Moreover, the authors indirectly suggest that the definition in [10] is unsuitable for K-way clustering, which is unclear. They state:

"However, they are not suitable for 𝐾-way clustering due to the uncertainty brought by the “the enemy of my enemy might be my enemy or my friend” principle of Weak Balance. We formally redefine positive and negative walks as follows."

I think I can understand the reasons behind their simplifications but it is not clear from the text. On the other hand, if the definition in [10] is indeed inappropriate for K-way clustering, this raises broader questions:

i. Balance in K-Way Clustering:
   Assume we have three communities. According to the modern definition of balance, such as that in [10], cycles with an even number of negative ties are balanced, while cycles with an odd number of negative ties are imbalanced. However, the authors only considered the cycles with zero negative ties as balanced. In K-way clustering, cycles with K negative ties (or any integer multiple of K negative ties) should also be recognized as balanced, based on their reasoning. For example for K=3, all cycles with 3 negative ties and integer multiple of 3 negative ties should be considered as balanced. What, then, is the treatment of other odd numbers of negative ties (e.g., 5, 7, 11)? Shouldn't these cycles be considered imbalanced? Please clarify.

ii. Cycles with More Than One Negative Tie:
   The authors appear to exclude cycles with an odd number of negative ties greater than one, potentially to align their method with K-way clustering (e.g., balancing cycles with three negative ties for three clusters). If so, what about cycles with an even number of negative ties? Are they considered balanced or imbalanced? For instance, one could construct:
   - A cycle with two negative ties between two communities that is balanced.
   - A cycle with two negative ties spanning three communities that is unbalanced.
These examples suggest that excluding cycles with more than one negative tie may lead to inconsistencies in the balance definition, especially when extended to K-way clustering. Could the authors clarify their reasoning for these choices and their implications on balance theory?

2. I think \alpha_l should be non-negative; however, based on the definition (4), the weight for \alpha_{L'} is negative. And more importantly, why do the authors need to define \alpha_{L'} this way? Please clarify. Why not just 1/L'! ?

3. Equations 11 and 12 do not make sense dimension-wise. \sigma(XW) is |V| \times d and W_1 is d \times d ? Please clarify and fix the notations to have the right equations. Possibly the authors meant Z^{+(0)} = \sigma(XW_0^+){W_1^+}^{T} ?


4. All the state-of-the-art methods evaluated for deep clustering in the paper are designed for unsigned graph clustering, and their deteriorating performance is not surprising. However, the authors could have included more relevant methods specifically designed for signed graphs. Examples include:

	Signed Spectral Embedding (SSE): A spectral clustering algorithm utilizing a signed version of the Laplacian matrix [J. Kunegis, S. Schmidt, A. Lommatzsch, J. Lerner, E. W. De Luca, and S. Albayrak, “Spectral analysis of signed graphs for clustering, prediction, and visualization,” Proceedings of the 2010 SIAM International Conference on Data Mining. SIAM, 2010, pp. 559–570].

	SiNE (Signed Network Embedding): A deep learning-based framework incorporating extended structural balance theory to embed signed graphs [S. Wang, J. Tang, C. Aggarwal, Y. Chang, and H. Liu, “Signed network embedding in social media,” Proceedings of the 2017 SIAM International Conference on Data Mining. SIAM, 2017, pp. 327–335].

	SIDE (SIgned Directed network Embedding): A random walk-based approach leveraging balance theory to capture indirect relationships in signed networks [J.-E. L. Junghwan Kim, Haekyu Park, and U. Kang, “SIDE: Representation learning in signed directed networks,” Proceedings of the 27th International Conference on World Wide Web (The Web Conference), 2018].

	Signed Embedding Using Graph Attention: A method for embedding signed graphs by applying graph attention mechanisms [Y. Li, Y. Tian, J. Zhang, and Y. Chang, “Learning signed network embedding via graph attention,” Proceedings of the AAAI Conference on Artificial Intelligence, Vol. 34, No. 4, 2020, pp. 4772–4779].

Including these methods---or other relevant approaches tailored for signed graphs---would provide a more comprehensive and fair evaluation of the proposed framework.


5. Some of the details are not explained well and that makes the interpretation of the results hard. Based on the SSBM, they create a stochastic block model with the probability of 𝑝 (edge probability or sparsity). I assume this probability is the probability of connections inside the communities. However, I expected there should be another probability of connections between the communities. Also, \eta is the probability of flipping the sign of the created edges. Possibly, in this model, the authors considered the same probability of connections within and between communities. However, the edges inside the communities are positive, and the edges between the communities are negative. And with the probability of \eta, they flip the sign of each edge. Please clarify.

6. Fig. 2's caption is not clear. Please make it clear. You say v3 is a negative neighbor of vi based on EEF, but I couldn't follow it. From what I can observe based on EEF (enemy of my enemy is my friend), I cannot conclude the relationship between vi and v3; however, using FFF, I can learn it is a positive neighbor?!

7. In line 804, Z_{eef}^+ = Z^+ + (\bar{A}^-)^2 Z^{+(0)} and Z_{eef}^- = Z^- why the added EEf part is only added to the Z^+ ? It is not clear based on the text. These details could be explained somewhere in the main text or the supplementary, but there are many ambiguities like this for the paper.

8. As I mentioned before, there is no ground truth theorem for community detection algorithms [Peel, L., Larremore, D. B., & Clauset, A. (2017). The ground truth about metadata and community detection in networks. Science advances, 3(5), e1602548.]. I am not sure if I can judge what Fig. 7 means visually. It is visible that there are clusters in both SPONGE and the DSGC method; however, it is not clear what we expect to see without ground truth. I just can say for real data this is not the right way to judge. A better approach to evaluate different clustering methods for real data is through indirect methods like link prediction [Ghasemian, A., Hosseinmardi, H., & Clauset, A. (2019). Evaluating overfit and underfit in models of network community structure. IEEE Transactions on Knowledge and Data Engineering, 32(9), 1722-1735.]. For example, by removing some percentage of the edges, you know the ground truth, and if your method could predict those removed links better, then I could be convinced that your method is working well. However, the results provided by the authors can only convince me that the results are good for the specific parameters of a synthetic model, nothing special for real data.

9. The proposed toy models' realizations assume that almost all positive edges are within communities and almost all negative edges are between communities (\eta < 0.04 in Fig. 4). Therefore, it makes sense that denoising improves the accuracy over these synthetic data. However, I am not convinced that the synthetic model's inferred parameter \hat{\eta} for a real network would be less than 0.04. In Fig. 4, it can be observed that around \eta = 0.04, the performance with and without VR-S is nearly the same, and for larger values, it even appears better without VR-S. This raises doubts about the applicability of the method to real networks. For example, weak balance theory supports the existence of negative ties within communities, and many real networks indeed exhibit negative ties inside communities.

**Reviewer Confidence:**

3: The reviewer is confident but not certain that the evaluation is correct

**Scope:**

3: The work is somewhat relevant to the Web and to the track, and is of narrow interest to a sub-community

---

### Official Review · Reviewer_Hcwh · 2024-12-03

**Novelty:** 6
**Technical Quality:** 6

**Review:**

The paper proposes the Deep Signed Graph Clustering (DSGC) framework, which addresses key challenges in signed graph clustering, particularly the vulnerability of existing spectral methods to noise and the limitations of Social Balance Theory when clustering multiple communities. It introduces the Deep Signed Graph Clustering framework by leveraging Weak Balance Theory.

Pros:
1. DSGC effectively uses Weak Balance Theory to improve both the preprocessing and encoding phases for more robust representation learning. It introduces two key strategies: Violation Sign-Refine to correct noisy edges, and Density-based Augmentation to enhance the graph structure.
2. Extensive experiments on both synthetic and real-world datasets demonstrate the effectiveness and robustness of DSGC.
3. The paper is well-written and easy to follow. It uses illustration figures and visualizations to effectively present the main ideas and results.

Cons:
1. The technical contribution is not very significant. The differences between the proposed DSGC solution and existing methods should be more clearly highlighted.
2. The paper lacks an analysis of time complexity or running time. Scalability is an important factor for the practical application of Signed Graph Clustering.

**Questions:**

This paper relies on Weak Balance Theory. Is it possible to incorporate other social theories into the proposed solution for more general signed social networks?

**Reviewer Confidence:**

2: The reviewer is willing to defend the evaluation, but it is likely that the reviewer did not understand parts of the paper

**Scope:**

3: The work is somewhat relevant to the Web and to the track, and is of narrow interest to a sub-community

---

### Official Review · Reviewer_kauj · 2024-12-03

**Novelty:** 6
**Technical Quality:** 5

**Review:**

The paper is devoted to the task of clustering of signed graphs. Authors note that current algorithms suffer from noisy edges and that Balance theory principle "an enemy of my enemy is my friend" should be eliminated for the case of K > 2 clusters. They suggest two techniques, Violation Sign-Refine (VS-R) which corrects noisy edges in a given signed graph, and Density-based augmentation which adds positive and negative edges based on neighbors at the distance more than 1. These techniques are incorporated in a signed graph convolution network, which forms an encoder, named DSGC. Embeddings obtained by the encoders are then clustered using soft clusters and optimisation. Experiments show that the suggested method outperforms SotA algorithms on synthetic graphs by accuracy, and gives better clusters on 2 real unlabeled graphs.
The strong point of the work is interesting and novel pre-processing techniques, which can be used besides the DSGC algorithm itself. Also, authors conduct comprehensive ablation study to show the influence of each technique and other details to the overall performance.
As a weakness of the work I see insufficient experiments on real graphs. Only 2 datasets are considered, and the results are evaluated only visually. No internal clustering metrics are considered (e.g. Silhouette Score) or stability metrics (like Cluster Consistency).

**Questions:**

There are some notes on the correctness and well-reading:
* On Figure 1, in a) or b), BNC and SPONGE seem to be mixed - each method should have same quality at x=0 on a) and on b)
* Figure 2 in introduction needs more explanation. Given only introduction, it is unclear what are Positive Feature Cluster and Negative Feature Cluster, whether $v_1$ and $v_4$ nodes are meaningful .
* The last coefficient in formula (4) is negative and goes to 2-e=-0.718 as $L'$ goes to infinity. Probably it should go to zero in the limit.
* According to (13), $\theta_k$ should have length $2d$ since it is multiplied by $Z_i$
* I would like the notation $C_{(:,𝑘 )} (𝑖)$ to be explained
* Numbers on some plots are too small (need to zoom a lot to read)

Questions to authors about the algorithm and results.
* The explanation of a signed graph convolution network in 4.2 was not enough for me. According to (9) and (10), $Z^{+(l)}$ and $Z^{-(l)}$ are determined, so the only trainable parameters are $w^+, w^-$. When are they learned? When the parameters from (11) and (12) are learned?
* Why did not you use any standard metrics for unlabeled clustering?
* From figure 4, it seems that VS-R and Regu do not contribute much to the performance of DSGC. However, 5.4 shows that VS-R has a great effect for other baseline algorithms. I would like to see some clarification about that.
* Hyperparameters 𝛿+, 𝛿 − , 𝑚+, and 𝑚− are chosen on 1 synthetic graph (as follows from App.C). But their sensitivity to graph is not explored, I suspect the optimal values can depend on graph properties.
* In the formulae (13), belonging to a cluster is scored as a projection of the embedding to $\theta_k$, so the vector $\theta_k$ of size 2d represents a cluster in a parameter space and ideally they form independent directions. Then it seems that if the number of clusters K > 2d, it leads to poor performance. Did you investigate that? What was $d$ value in the experiments?

**Reviewer Confidence:**

3: The reviewer is confident but not certain that the evaluation is correct

**Scope:**

4: The work is relevant to the Web and to the track, and is of broad interest to the community